# Learning about climate change uncertainty enables flexible water infrastructure planning

Sarah Fletcher [1,2], Megan Lickley [3] & Kenneth Strzepek[2]

Water resources planning requires decision-making about infrastructure development under uncertainty in future regional climate conditions. However, uncertainty in climate change projections will evolve over the 100-year lifetime of a dam as new climate observations become available. Flexible strategies in which infrastructure is proactively designed to be changed in the future have the potential to meet water supply needs without expensive over-building. Evaluating tradeoffs between flexible and traditional static planning approaches requires extension of current paradigms for planning under climate change uncertainty which do not assess opportunities to reduce uncertainty in the future. We develop a new planning framework that assesses the potential to learn about regional climate change over time and therefore evaluates the appropriateness of flexible approaches today. We demonstrate it on a reservoir planning problem in Mombasa, Kenya. This approach identifies opportunities to reliably use incremental approaches, enabling adaptation investments to reach more vulnerable communities with fewer resources.

[1] Department of Civil and Environmental Engineering, Massachusetts Institute of Technology, 77 Massachusetts Avenue, 48-216, Cambridge, MA 02139, USA. [2] Joint Program on the Science and Policy of Global Change, Massachusetts Institute of Technology, 77 Massachusetts Avenue, E19-411, Cambridge, MA 02139, USA. [3] Department of Earth, Atmospheric, and Planetary Sciences, Massachusetts Institute of Technology, 77 Massachusetts Avenue, 54-1713, Cambridge, MA 02139, USA. Correspondence and requests for materials should be addressed to S.F. (email: sfletch@mit.edu)

Uncertainty in climate change projections poses a challenge to infrastructure planning for climate change adaptation[1]. Because of the large expense and widespread need for adaptation investments, planning models play a critical role in targeting resources. Traditional water infrastructure planning accounts for uncertainty by adding a safety factor to new infrastructure[2]. However, these large projects are typically irreversible, expensive, and last for multiple decades; the same is true across many infrastructure domains[3]. Preparing for climate change by adding extra capacity, therefore, incurs high risk of expensive overbuilding in resource-scare areas. Flexible infrastructure planning has the potential to manage uncertainty at reduced cost by building less infrastructure up front but enabling expansion in the future if needed[2,4,5]. However, enabling flexibility often requires substantial proactive planning or upfront investment[6]. In water resources, it is difficult to know whether recent trends in streamflow are a result of climate change or short-tern variability and therefore whether they are predictive of future trends[7]. It is therefore difficult for planners to know if and when to trigger adaptive actions. Short-term reliability outages can occur if infrastructure cannot be adapted quickly[8]. Further, flexibility can ultimately be more expensive by not taking advantage of economies of scale[6]. Appropriate methods are therefore needed to weigh the risks and benefits of static vs. flexible infrastructure approaches in responding to climate change uncertainty.

Several recent studies provide methods to develop and assess flexible (also called adaptive) infrastructure planning under climate change uncertainty. Robust decision making (RDM) uses iterative scenario development to minimize the regret from both overbuilding unnecessary infrastructure and being unprepared[9–11]. RDM has been used to develop and evaluate adaptive infrastructure planning strategies[12–14]. New policy-making processes design adaptive pathways that allow planners to switch from one action to another if specified thresholds are reached[15] and can be combined with optimization approaches to identify adaptive thresholds and actions[16]. Recent approaches have provided methods for adaptive sequencing of infrastructure investments[8,17]. Finally, advances in search algorithms[18,19] have enabled assessment of adaptive and cooperative approaches against many performance measures using ensembles of streamflow projections[20].

Adaptive management requires an ability to learn over time as more information is collected[5]. A challenge faced by the aforementioned approaches is the difficulty in assessing opportunities to learn in the future. General circulation model (GCM, i.e. climate model) projections provide us with the best available estimates of how the global climate system will evolve under a given emissions scenario. However, as time passes and new climate observations are available, some GCM trajectories will prove to be more reliable than others. For example, suppose current regional projections estimate a range between 0.5 and 1.5 °C of change over the next 20 years. If after 20 years we observe 1.5 °C of change, this suggests the climate is warming in this region more rapidly than expected. We may now shift our projections of change upward for the following 20 years. While existing frameworks provide an iterative process for planners to change course in the future, they do not provide an upfront assessment of the opportunity to learn about climate change in the future. This upfront assessment is critical to deciding whether investments in flexibility are worthwhile or whether a traditional static approach is more appropriate. Existing flexible approaches either assume a priori that flexibility is needed[8], assume perfect information about the future[21], or rely on thresholds or signposts that are unrelated to learning about climate change[13], but do not provide a mechanism for assessing opportunities to learn about climate change in the future. Recent studies have incorporated learning

feedback from short-term nonstationary streamflow, but not long-term climate change[13,22,23]. Note that while this study focuses on water supply infrastructure, the challenge of characterizing learning about climate uncertainty to enable adaptive planning has been highlighted in a range of other disciplines (for example in forest management[24]).

We develop a planning framework that explicitly models the potential to learn about climate uncertainty over time and uses potential learning to develop and evaluate flexible planning strategies in comparison to static approaches. First, we use GCM projections to develop a wide range of possible future mean regional temperature ($T$) and precipitation ($P$) outcomes over a planning horizon. We finely discretize mean annual $T$ and $P$ within that range. This develops a comprehensive set of virtual climate observations of mean $T$ and $P$ that reflect many possible future regional climates, some of which are drier and some of which are wetter. Next, we adapt a Bayesian statistical model[25] to update initial climate uncertainty estimates for each virtual climate observation. The updated estimates reflect what we will have learned if the virtual observation comes to pass. These updated uncertainty estimates characterize the transition probabilities in a non-stationary stochastic dynamic program (SDP); each possible change in SDP climate state is equivalent to a virtual climate observation. This SDP planning formulation therefore takes into account all the potential new information that may be learned in the future as it develops optimal planning policies. We use these polices to evaluate flexible infrastructure planning approaches and compare them to static approaches.

The United Nations Environment Program estimates that the cost of climate change adaptation investments in the developing world may reach $500 billion per year by 2050;[26] the World Bank estimates that the infrastructure and water sector adaptation costs may be $28 billion and $20 billion per year, respectively[27]. It is therefore essential to target infrastructure investments efficiently to reach the widest number of vulnerable communities. Flexible planning strategies can substantially reduce the cost of infrastructure investments. To the authors' knowledge, this is the first framework that values the ability of flexible approaches to respond to climate learning, therefore more comprehensively evaluating the tradeoffs of robust and flexible adaptation strategies.

Results show that climate change uncertainty can be reduced over the lifetime of an infrastructure project across different climate change trajectories. Flexibility is effective in preventing unnecessary infrastructure additions while maintaining similar reliability. However, the planning choice is informed by the social context including value of reliability and discount rate.

## Results

**Planning framework and scenarios.** We demonstrate this planning framework, illustrated in Fig. 1, with an application for Mombasa, Kenya. Mombasa is the second largest city in Kenya with an estimated population of 1.1 million[28]. Urban water demand is currently estimated at 150,000 m$^3$day$^{-1}$ and expected to grow to 300,000 m$^3$day$^{-1}$ by 2035[29]. Mombasa has a warm, humid climate with average annual precipitation of 900 mm year$^{-1}$ and a mean annual temperature of 26 °C[30]. Mean annual runoff (MAR) in the nearby Mwache river, the site of a proposed dam, is 113 MCM year$^{-1}$ [31]. While GCMs all project warming in the region, there is disagreement on the direction of precipitation change. This creates substantial uncertainty in future runoff and therefore the reservoir capacity needed to meet yield targets over its lifetime. We apply our framework to develop and assess a flexible infrastructure design. The flexible design enables extra

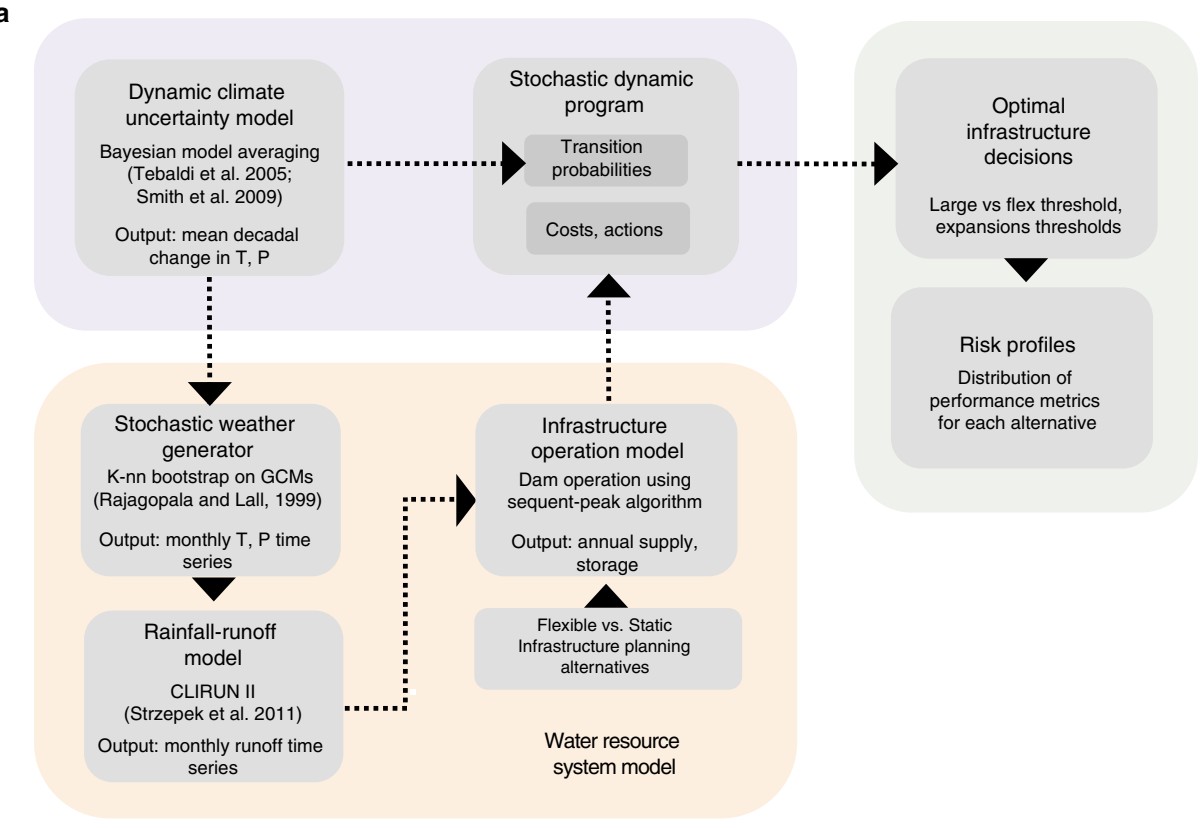

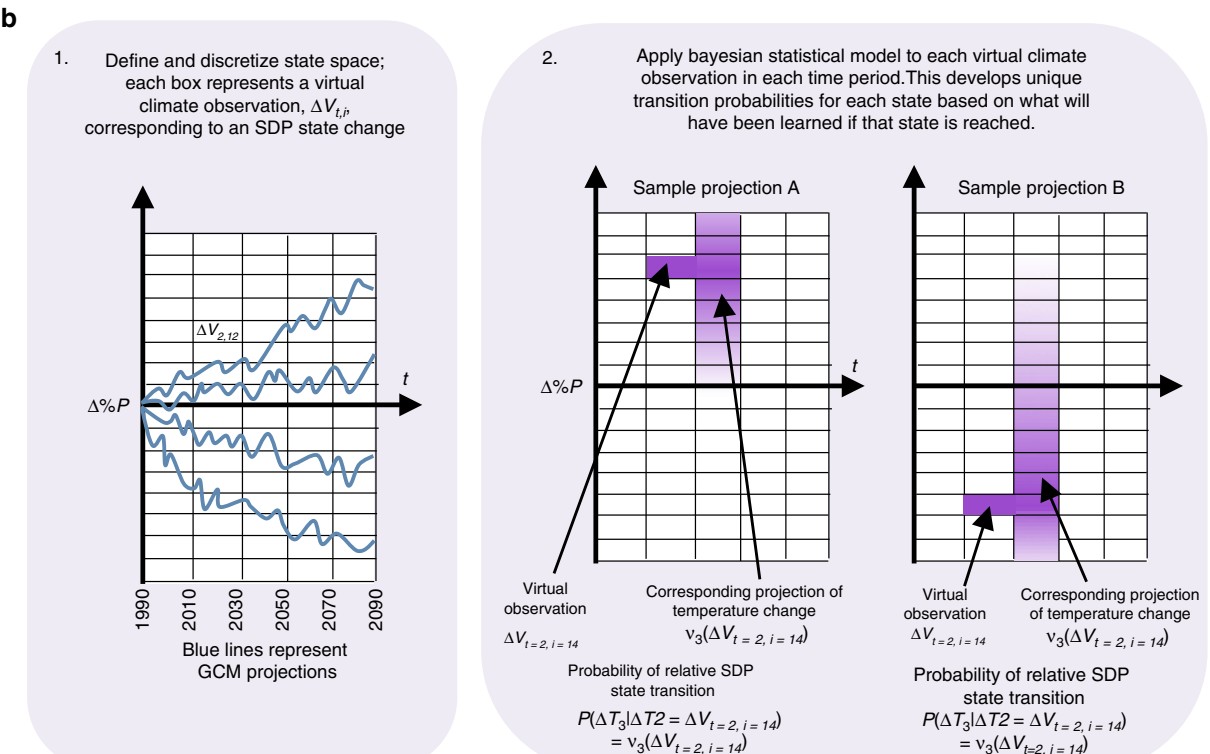

**Fig. 1** Schematic of integrated modeling framework. **a** Full planning framework. **b** Detail on characterizing transition probabilities using Bayesian statistical model applied to each virtual climate observation

**Table 1 Planning scenario definitions**

| Planning scenario | | Technology | DR | Capacity [MCM] | | | | Capex [M$] | |
|---|---|---|---|---|---|---|---|---|---|
| | Demand | | | Small | Large | Small | Large | Exp | Flex + Exp |
| A | Low | Earthen dam | 3% | 80 | 120 | 76.5 | 99.2 | 49.6 | 148.8 |
| B | Low | Earthen dam | 0% | 80 | 120 | 76.5 | 99.2 | 49.6 | 148.8 |
| C | High | RO desalination | 0% | 60 | 80 | 183.1 | 232.2 | 72.4 | 255.5 |

DR discount rate, RO reverse osmosis, Capex capital expenditure

storage capacity to be added if the initial dam becomes insufficient due to warmer, drier climates.

We assess three planning scenarios, described in Table 1, intended to evaluate the sensitivity of our results to social and technological planning assumptions. In the low-demand scenarios (A and B), we assume a target yield of 150,000 m³ day⁻¹ (54.8 MCM year⁻¹) with 90% reliability from the Mwache dam. We evaluate the two dam sizes proposed by the previous World Bank study[21], 80 MCM and 120 MCM, as well as a flexible alternative in which the height of the smaller dam can be raised, increasing the reservoir capacity to 120 MCM. In planning scenario C we assume a target yield of 300,000 m³ day⁻¹ (109.6 MCM year⁻¹) with 90% reliability over the entire planning horizon, reflecting the potential for rapid demand growth on relatively short timescales based on 2035 projections from[29].

In this scenario, the target yield is greater than observed MAR in the Mwache river, and therefore the dam cannot meet the target yield in today's climate regardless of its size. Therefore, we model the combination of a 120 MCM dam and a desalination plant that is used to supply demand when reservoir storage is low. Three desalination alternatives are chosen, analogous to the dam design alternatives. A low capacity alternative designed to meet reliability targets in the current and expected future climate with 60 MCM capacity; the large alternative that meets the reliability targets across all projected future climates with 80 MCM capacity; a flexible alternative starts with 60 MCM and can be expanded to 80 MCM. Evaluating this second scenario allows us to compare the value of flexibility across two technology options, earthen dams and desalination, which have unique water supply profiles and cost structures.

**Bayesian learning about climate change uncertainty**. Figure 2a, b show historical observed regional annual $T$ and $P$ from the Climate Research Unit (CRU)[32], as well as individual GCMs' projected changes in $T$ and $P$ relative to 1990. 90% confidence intervals (CIs) of GCM projections are developed using the Bayesian uncertainty approach, assuming the historical period is prior to 1990, and compared to CIs developed using a traditional democratic weighting. The Bayesian approach weights models based on how well they match historical observed changes in $T$ and $P$ (see Methods). The democratic approach assumes all models perform equally well[33]. Between these two methods, the Bayesian approach produces smaller CIs because it assigns more weight to a subset of models that best match historical change in this region.

While Fig. 2 presents Bayesian CIs based on historical observations, the SDP transition probabilities require Bayesian uncertainty estimates that reflect what will have been learned for many possible virtual future observations. We assume that precipitation change will range between −30% and +30% by end of century; we discretize this range at 2% for a total of 31 unique virtual precipitation change observations. We apply the Bayesian uncertainty analysis to each of these 31 virtual

precipitation change observations in each time period. For example, two sample time series of virtual $T$ and $P$ observations and their corresponding updated uncertainty estimates are shown in Fig. 3. An example of strongly increasing $P$ is shown at top; an example of modestly decreasing $P$ is at bottom. For each virtual observation, we simulate 10,000 virtual climate time series from the current observation to the end of the planning period and construct a 90% CI, shown by the shaded regions. This process is repeated for each time step, with darker colors in the plot corresponding to the CIs developed from virtual observations sampled later in the planning period. The darker CIs therefore reflect uncertainty estimates updated with information farther into the future. The sample of virtual observations showing strong increases in $P$ (Fig. 3a–d), leads to high certainty by the end of the century that negligible water shortages will be incurred, assuming the small 80 MCM of dam capacity. Strong asymmetric uncertainty reflects the low-probability, high-severity risk of droughts; shortages occur only when runoff is substantially below MAR for several months. The alternate sample of virtual observations showing modest decreases in $P$ (Fig. 3e–h) demonstrates a reduction in uncertainty in both $P$ and MAR. Expected water shortages increase substantially as more observations are collected, and the uncertainty increases as well due to non-linear relationships between MAR and shortages.

While two sample time series of observations are illustrated in Fig. 3, the SDP optimal strategy accounts for a wide range of possible future observations and what would be learned if they were to be observed. This is achieved through the multistage stochastic optimization formulation, which allows for uncertain, rather than deterministic, transitions to new climate states in each period. In the first time period, shown in Fig. 4a, the SDP develops a threshold as a function of $T$ and $P$ during the 2001–2020 time period when the initial infrastructure decision is made. Above the threshold, in hotter and drier climates, the large dam is optimal and below it the flexible dam is. Due to the small cost difference between the flexible and large dam, investing in the large dam option upfront is preferred if the risk of shortages at the outset is high enough. This reduces expected costs by leveraging economies of scale. Panel b shows expansion thresholds for time periods 2–5 for the flexible dam. Expanding infrastructure capacity is optimal in drier and warmer states. In the 2041–2060 time period, the policy threshold shifts right, reflecting the narrowing of uncertainty due to additional information in later time periods. In later time periods, however, it shifts left, reflecting the influence of the end of the planning horizon which disincentivizes investment.

Figure 5 shows infrastructure decisions under the optimal policy across 1000 simulated climate time series. In planning scenario A, the flexible alternative is chosen in 90% of simulations, shown in panel a. When the flexible alternative is chosen, the option to expand is never chosen in about 90% of simulations. This highlights the low probability of reaching a climate dry enough to generate shortages beyond 10% of demand. The time period at which expansion is exercised varies; more

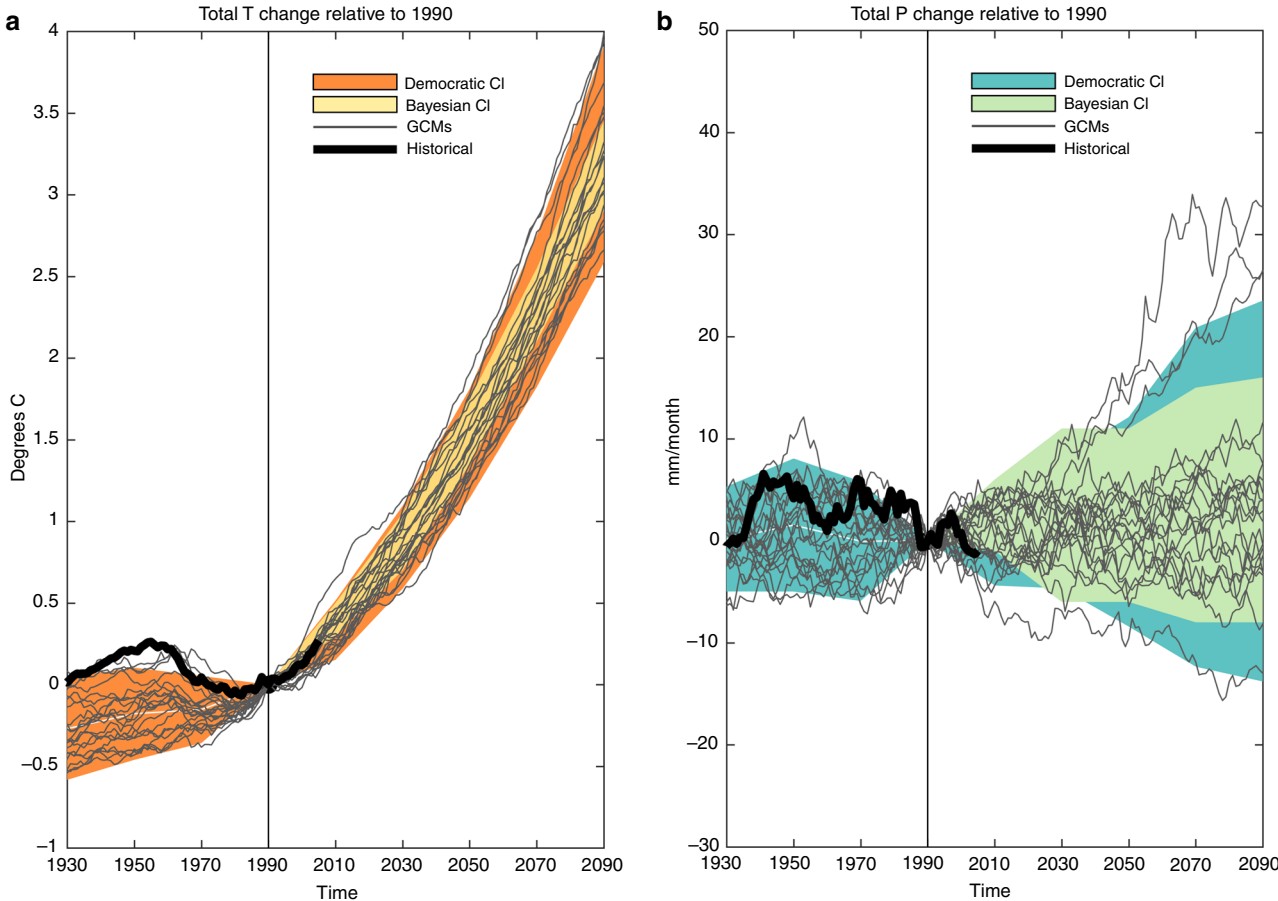

**Fig. 2** Bayesian and democratic confidence intervals from GCM projections. **a** and **b** Modeled and observed temperature (precipitation) relative to 1990 values with uncertainty estimates. Thin gray lines are 20-year moving averages of GCM simulations over Mombasa. Thick black lines show the corresponding historical observed values. Orange (blue) shaded regions show the 90% CIs using the IPCC democratic weighting method (i.e. $\pm1.64*\sigma$). Yellow (green) shaded regions show the 90% CI developed using the Bayesian uncertainty method applied to historical regional observations before 1990

rapid warming and drying leads to earlier expansion. Panel b shows cumulative distribution functions (CDFs) of the total cost (including shortage damages) of each alternative across the 1000 simulations under planning scenario A. The large static alternative has the same cost across simulations; as designed, no shortage damages are incurred in any feasible climate. The small dam performs better than the large dam in about 70% of simulations, but has substantially higher costs in 30% of simulations due to large damages from water shortages. The flexible dam mirrors the small dam in 70% of simulations, but the reliability risk is substantially mitigated because of the potential to expand. The high-end costs are higher than the large dam because, first, the cost of building the 80 MCM dam and expanding to 120 MCM is higher than building the 120 MCM dam upfront and, second, sometimes the dam is not expanded even when modest water shortages are incurred. The ability of the flexible alternative to mitigate both the risk of overbuilding and the risk of severe shortages demonstrates the high value of flexibility in this case.

The value of flexibility changes under planning scenarios B (no discounting; panels c and d) and C (high demand with desalination plant; panels e and f). Without discounting, the large dam is more favorable; it performs best in 60% of simulations, has no cost variability risk, and is chosen in 80% of simulations. Large economies of scale in the dam mean that a 120 MCM dam is only 30% more expensive than an 80 MCM dam for 50% additional capacity. This suggests it is often better to

build the large dam upfront even if there is a relatively low probability that it will be needed. Scenario C evaluates a 120 MCM dam combined with a desalination plant. We find a high value of flexibility even without discounting. The flexible alternative is chosen upfront in over 80% of forward simulations. The CDF demonstrates that it outperforms the static alternatives by substantially mitigating the over build risk in comparison to the robust alternative. The flexible alternative also modestly reduces the shortage damage risk in comparison to the small alternative. While the flexible alternative only reduces cost at the 90th percentile and above, this substantially reduces the expected value as the maximum cost of the small plant reaches almost M $400.

Looking across scenarios, the flexible alternative is chosen most often in scenario A because discounting incentivizes delayed capital investments. This is not the case in scenario B because large economies of scale incentivize a single, large investment. In scenario C more modest economies of scale lead to high value of flexibility in the absence of discounting, highlighting differences in the value of flexibility across technologies. Across all scenarios, the flexible dam is expanded in no more than 10% of simulations, highlighting the low probability of reaching a climate that is hot and dry enough to incur substantial shortages.

Finally, while the previous analysis has relied on a top-down analysis that uses GCM projections to develop probabilistic forecast, Fig. 6 presents an illustrative bottom-up analysis that demonstrates the average cost and regret of each of the three dam

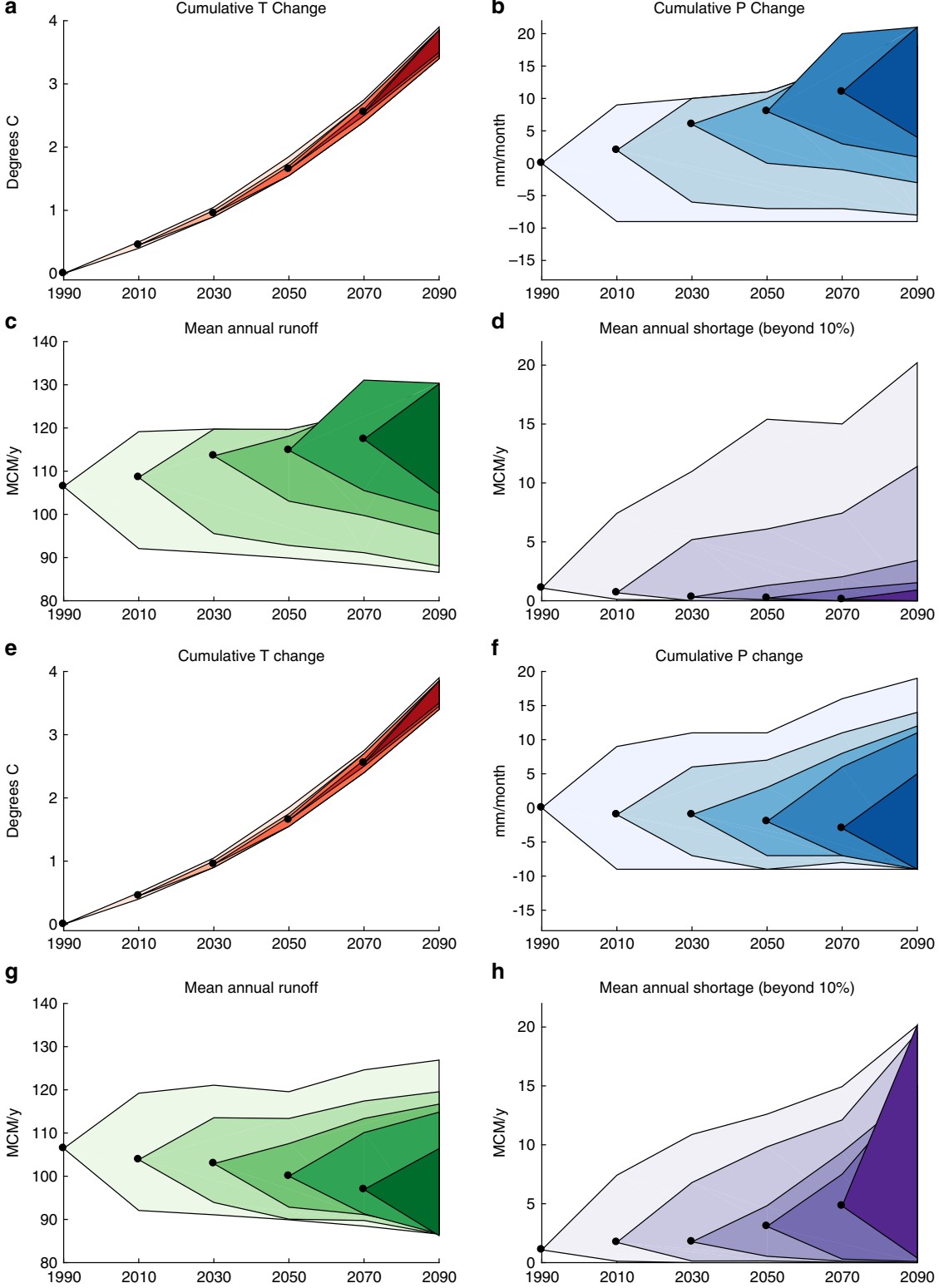

**Fig. 3** Learning over time using virtual observations. **a–d** One sample realization of Bayesian learning over time in which precipitation increases strongly. Black dots represent a time series of virtual climate observations. Shaded regions indicate the projected 90% CI, updated with each time period's virtual observation. Virtual observations of *T* (**a**) and *P* (**b**) are used to simulate MAR (**c**), and water shortages assuming 80 MCM dam capacity (**d**). **e**, **h** As in **a–d** but for an alternative realization of virtual observations, showing modest decrease in *P*

alternatives in planning scenario A under different end-of-century climates without relying on probabilistic forecasts. Regret is defined as the difference between the cost of the chosen infrastructure alternative and the best possible infrastructure

alternative in a given climate state. Three illustrative climates are chosen to demonstrate the tradeoffs across alternatives: a dry climate of 68 mm month$^{-1}$, an moderate climate of 78 mm month$^{-1}$, and a wet climate of 88 mm month$^{-1}$. Differences in

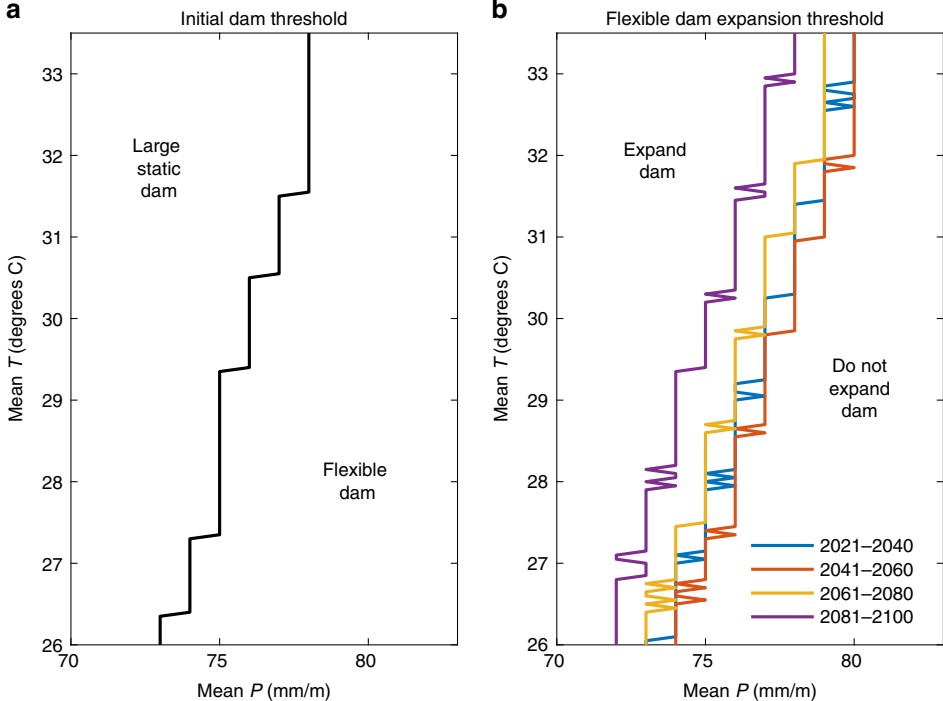

**Fig. 4** Optimal policies from SDP. **a** Threshold for initial decision between large static and flexible design as a function of $T$ and $P$ during the first time period (2000–2020). **b** Thresholds for exercising the option to increase height of flexible dam as a function of $T$ and $P$ during the latter time periods as indicated on the legend. Results shown for planning scenario A

$T$ are not considered because its impact on water shortages is limited. The small dam without expansion has the highest maximum regret of any alternative of M\$77, incurred in the dry climate. The large dam incurs positive regret in both the moderate and wet climates, with the latter incurring M\$19 of regret. The flexible dam has the lowest maximum regret, with a modest M\$4 of regret in the dry climate. This bottom up approach also highlights the ability of the flexible dam design and expansion strategy to mitigate risk in a range of different potential future climates.

## Discussion

We develop a method that integrates iterative Bayesian learning about climate uncertainty into a multi-stage stochastic infrastructure planning model in order to address a critical limitation of adaptive infrastructure planning in both water supply and other domains: estimating upfront how much planners can expect to learn about climate change in the future and therefore whether adaptive approaches are likely to be reliable and cost effective. Our approach quantifies, for example, the extent to which a wet trajectory over the next 20 years increases the likelihood of a wet trajectory 40 years into the future. By applying the Bayesian model to a wide range of discrete virtual future climate observations, we develop adaptive policies that take into account all future opportunities for learning. While all approaches that use GCM ensembles face limitations, this approach provides a reasonable quantitative estimate of future learning that enables better-informed assessment of tradeoffs between planning approaches. This allows us to evaluate the effectiveness of flexible planning, which relies on learning processes that remain unquantified in previous methods, rather than assuming a priori that flexibility is a worthwhile planning goal. This is especially important for infrastructure planning where planners must prepare in advance to take a flexible approach due to the large, irreversible nature of infrastructure investments.

The results in the Mombasa application demonstrate the nuances and tradeoffs inherent in comparing flexible and robust approaches for planning under climate uncertainty. Although the uncertainty and learning is driven by the climate system, decisions about whether flexibility is a valuable tool in mitigating risk are strongly influenced by social, technological, and economic factors. The large economies of scale in earthen dams make flexibility less valuable; it is better to choose a robust alternative when it is not much more expensive to do so. Reverse osmosis (RO) desalination, however, is an inherent modular technology with modest economies of scale, lending itself more readily to flexible planning. The discount rate, which trades off future adaptation goals for immediate rewards, promotes flexible approaches. Flexibility often delays investment, which can be especially impactful in resource-scarce areas where unused capital could support other critical infrastructure services. The value society places on access to reliable, sustainable water supplies, and the damage of short-term outages is also influential.

Future extensions to other planning problems which have differences in degree and nature of uncertainty, hydrological sensitivity to climate change, and social context can be used to assess under what conditions flexible or static planning approaches are more appropriate. Future work combining this learning approach with bottom-up vulnerability assessments can address the limitations of GCM-based probability distributions[34]. This framework shows promise in identifying areas where smaller, flexible infrastructure is reliable vs. those that require a traditional static approach, enabling billions of dollars of potential savings in climate change adaptation investments across civil infrastructure domains.

## Methods

**Bayesian modeling of climate change uncertainty**. We extend previous Bayesian uncertainty analysis of climate change[25] (hereafter called previous Bayesian model) to characterize the SDP transition probabilities. Previous work shows that the uncertainty in climate projections due to natural variability remains relatively

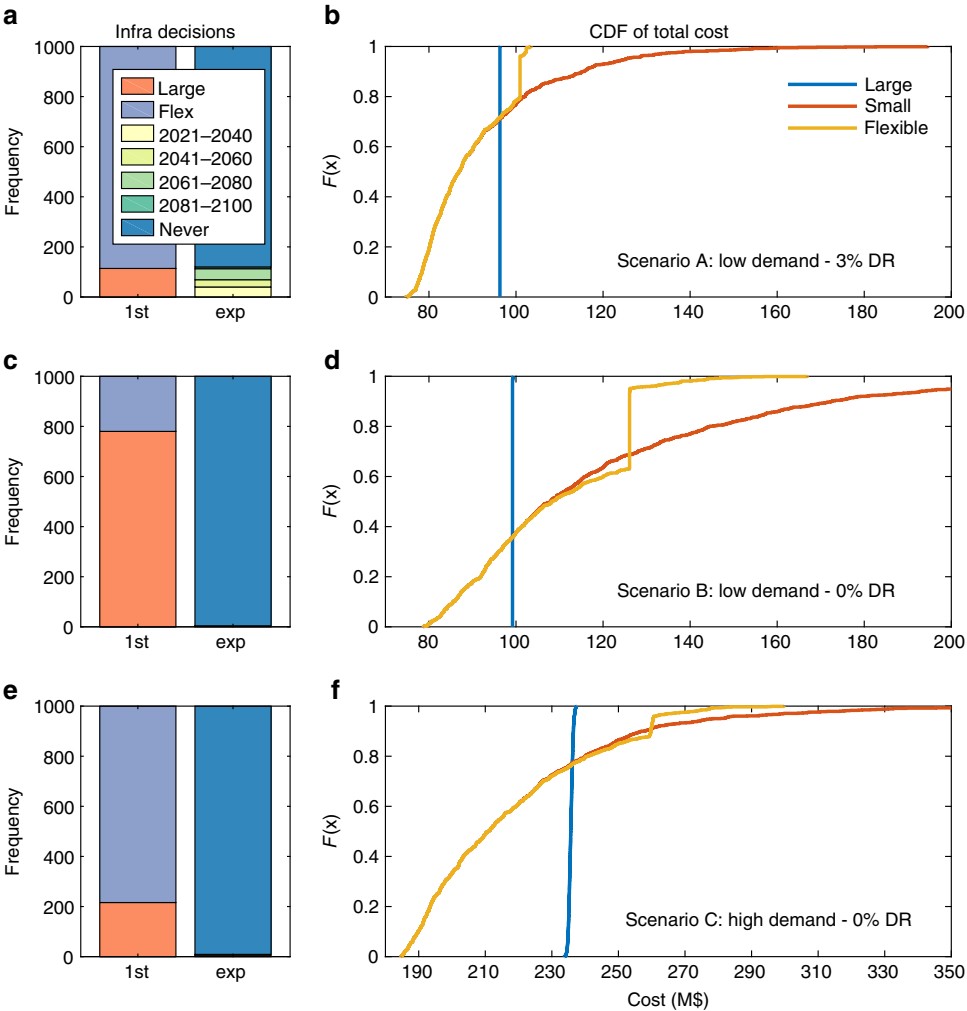

**Fig. 5** Simulated infrastructure decisions and costs. **a**, **b** Planning scenario A (low-demand, discounting). **c**, **d** planning scenario B (low-demand, no discounting); **e**, **f** planning scenario C (high-demand, no discounting

constant throughout the 21st century, but that as the climate signal emerges from the noise, the uncertainty in projections is dominated by the GCMs' climate sensitivity, and hence structure[35]. We therefore limit our focus to uncertainty in model structure rather than emissions or stochasticity because, first, structural uncertainty dominates long-term precipitation uncertainty[35] and, second, to utilize recent statistical methods for characterizing structural climate uncertainty[25,36]. The previous Bayesian model[25] uses ensembles of projections from the fifth phase of the Coupled Model Intercomparison Project (CMIP5)[37] to derive a single distribution describing uncertainty in climate change. Following the previous Bayesian model[25], we use historical observations or virtual observations to estimate the reliability of each model run and therefore its weight in the resulting probability distribution. This is in contrast to the democratic approach[38] which each model projection is assumed equally likely and the multi-model mean and standard deviation is used to derive a single probability distribution.

We extend the previous Bayesian model[25] in three ways. First, we apply the model to annually averaged $P$ and $T$ values separately, assuming that $T$ and $P$ are independent. This reflects that a model's performance in estimating $T$ may be unrelated to its ability to estimate $P$. Second, we apply the model to observed and projected change in $T$ and $P$ (i.e. $\Delta T$ and $\%\Delta P$) rather than absolute $T$ and $P$ due to greater model skill in GCM projected changes in temperature and precipitation rather than absolute values[39,40]. This is especially important in our application in Mombasa where there is less disagreement in temperature change than there is disagreement in hind-casted absolute temperature.

Finally, we apply the model to multiple pairs of time windows and also to many virtual observations of change in $T$ and percentage change in $P$. The previous Bayesian model[25] assumed two periods: a historical climate (1961–1990) and a future climate (2071–2100). We also use a historical and future climate in each estimation of the Bayesian model; however, we define 6 time periods using pairs of adjacent 20-year windows and calculate the change in $T$ and percentage change in $P$ between adjacent windows. This gives a total of five pairs of historical and future adjacent windows within 1960–2099. In each pair of adjacent windows, the

historical window corresponds to the current time period in the SDP and the future window corresponds to the next 20-year period; this is necessary for the 1-stage transition probabilities needed in the SDP. The 20-year time interval was chosen so that interannual variability was not driving the trend in precipitation and temperature across time periods. The previous Bayesian model[25] used historical observations of climate data ($X_0$ in (Eq. 1)); we repeat the analysis many times using unique virtual climate observations, $\Delta V_{t,i}$, corresponding to changes in the SDP climate states, where $t$ denotes the time period and $i$ denotes an index between 1 and $N$, the possible virtual observations. Virtual temperature change observations range from 0 to 1.5 °C using discrete steps of 0.05 °C ($N = 31$). Virtual observations of percentage change in precipitation range from −30% to 30% using discrete steps of 2% ($N = 30$). These were chosen in order to be comprehensive of all potential future climate states. Therefore, they must be, first, granular enough that adjacent observations result in similar distributions and therefore approximate a continuous set of observations and, second, span a range that exceeds the full range of change predicted by models (i.e. a range of 0–1.5 °C per 20-years is equivalent to 0–7.5 °C of change after 100 years; the CMIP5 ensemble projections a temperature change in the range of 2–4 °C by 2100, fitting well within the range resulting from the virtual observations).

The evaluation of GCMs' performance in reproducing climate observations will depend on time scale, region, and variable of interest[41,42]. Because our ultimate goal is to update our learning of regional climate in the Mwache catchment with respect to multi-decadal trends in precipitation and temperature, we choose to weight GCMs based on their performance in reproducing multi-decadal trends of precipitation and temperature averaged over the catchment area. Therefore, to implement the Bayesian uncertainty analysis in Mombasa, we use a total of 21 CMIP5 members whose modeling group and model run are included in Supplementary Table 1. The 21 GCM simulations come from 10 different institutions and 15 different GCMs, with three GCMs providing more than one simulation. Models were selected based on the most readily available models at the time of the analysis, with 21 being in line with previous studies, providing a

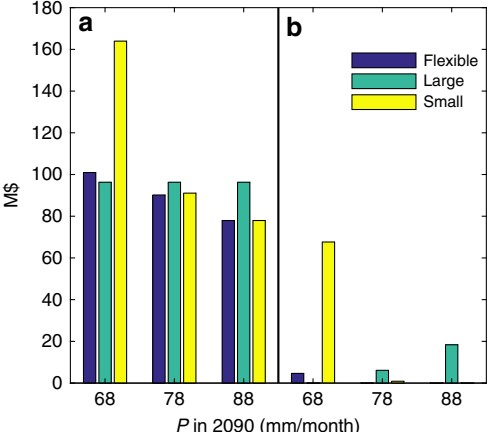

**Fig. 6** Illustrative bottom-up regret analysis. **a** Total cost including shortage penalties; **b** regret by infrastructure alternative in planning scenario A. Cost and regret are assessed in three representative end-of-century $P$-values: a dry climate of 68 mm month$^{-1}$, a moderate climate of 78 mm month$^{-1}$, and a wet climate of 88 mm month$^{-1}$

reasonable balance between computational limits and model diversity[33]. All models are forced by the RCP 8.5 scenario, which is the high emissions scenario from the IPCC AR5. For each GCM, monthly temperature and precipitation values are averaged over 2–6°S and 38–42°E, overlaying the Mwache catchment; GCM projections are regridded from their original resolution following the approach in[43]. These regional temperature and precipitation GCM outputs, rather than global outputs, provide the basis for model weighting in the Bayesian analysis.

Our statistical model is formulated as follows for $\Delta T$; an identical and independent model is used for $\%\Delta P$. The estimate of future change in mean temperature between $t=0$ and $t=1$, $v_1$, is based on historical observed temperature change to $t=0$, $\Delta X_0$:

$$\Delta X_0 \sim N(\mu_0, \lambda_0^{-1})$$
$$\Delta X_0^j \sim N(\mu_0, \lambda_0^{j-1})$$
$$\Delta X_1^j | \Delta X_0^j \sim N(v_1 + \beta_0 * (\Delta X_0^j - \mu_0), (\theta_0 * \lambda_0^j)^{-1}) \quad (1)$$

where $\Delta X_0$ is the historical observed temperature change to $t=0$. $\Delta X_0^j$ is model $j$'s projection of temperature change to $t=0$, and $\Delta X_1^j$ is the same for $t=1$. $\Delta X_0$, $\Delta X_0^j$, and $\Delta X_1^j$ are treated as samples from unique normal distributions. $\mu_0$ and $v_1$ are random variables representing the underlying distributions of temperature change in the current ($t=0$) and future ($t=1$) time periods respectively. $\lambda_0^j$ is the inverse variance of $\Delta X_0^j$, representing the reliability of model $j$. $\beta_0$ is a regression parameter that introduces correlation between $\Delta X_0^j$ and $\Delta X_1^j$; it is estimated by the model rather than assumed. $\theta_0$ is also an estimated parameter that enables a model to have different reliability in the future compared to the present. The marginal densities for each of the parameters are estimated using MCMC methods; we use the Gibbs sampling approach, parametric assumptions including priors, and code developed in ref. [25]. The Gibbs sampler collected 1000 samples, discarded the first 150,000 samples as a burn-in, and saved 1 in every 1500 samples; convergence was checked using standard diagnostics including trace plots and auto-correlation plots.

When $t>1$, unique estimates of future change in mean temperature from $t-1$ to $t$, $v(\Delta V_{t-1,i})$, are based on each virtual observation of temperature change from the previous time period, $\Delta V_{t-1,i}$, as follows:

$$\Delta V_{t-1,i} \sim N(\mu(\Delta V_{t-1,i}), \lambda(\Delta V_{t-1,i})^{-1})$$
$$\Delta X_t^j \sim N(\mu(\Delta V_{t-1,i}), \lambda^j(\Delta V_{t-1,i})^{-1})$$
$$\Delta X_t^j | \Delta X_{t-1}^j \sim N(v(\Delta V_{t-1,i}) + \beta(\Delta V_{t-1,i}) * (\Delta X_{t-1}^j - \mu(\Delta V_{t-1,i})), [\theta(\Delta V_{t-1,i}) * \lambda^j(\Delta V_{t-1,i})]^{-1})$$
$$\forall i = 1, \dots, N; \ t = 2, \dots, 5 \quad (2)$$

where the notation is analogous to that in Eq. (1) except that now $N$ unique distributions are estimated corresponding to each virtual observation. Virtual observation $\Delta V_{t-1,i}$ is treated as a sample from an underlying normal distribution; $\mu(\Delta V_{t-1,i})$ and $v(\Delta V_{t-1,i})$ are the underlying change in mean temperature in the current ($t-1$) and future ($t$) time periods, respectively, given each virtual observation $\Delta V_{t-1,i}$; $\lambda^j(\Delta V_{t-1,i})$ is the reliability of model $j$ for virtual observation $i$ in time $t$; and $\beta(\Delta V_{t-1,i})$ and $\theta(\Delta V_{t-1,i})$ are estimated uniquely for each virtual observation $\Delta V_{t-1,i}$.

This approach does have limitations. First, it assumes that GCMs are independent of one another, when in fact some models borrow entire components from other models[44]. Second, we assume that a GCM's ability to reproduce $\Delta T$ or $\%\Delta P$ is a better indication of model performance than another metric, such as model variability. Third, we assume that change in time $t$ depends on $t-1$ and not previous time periods. Additionally, we assume climate models will not change in the future; repeating the analysis in 40 years with a broader range of models

reflecting the new state of the science may produce larger shifts in CIs. However, this approach is the best available to estimate learning in the future, which impacts planning decisions today. It enables a more precise measure of uncertainty in comparison to the democratic approach used by the IPCC; it has also been statistically validated using a cross validation approach[25].

**Estimating transition probabilities.** Each estimate for $v(\Delta V_{t-1,i})$ (or $v_1$ if $t=1$) is then used to estimate the probability of change in each temperature state $T_t$ in the SDP temperature state space $\mathbf{S}^T$. (Note we treat $v(\Delta V_{t-1,i})$ as a probability mass function discretized at the same granularity as the virtual observations):

$$P(\Delta T_t | \Delta T_{t-1} = \Delta V_{t-1}) = v(\Delta V_{t-1,i})$$
$$P(\Delta T_t = a | \Delta T_{t-1} = \Delta V_{t-1}) = P(v(\Delta V_{t-1,i}) = a) \quad (3)$$
$$\forall i = 1, \dots, N; \ t = 1, \dots, 5$$

We then define the joint distribution for the relative change probabilities using the chain rule and the Markov assumption, which is consistent with our assumption in the Bayesian model that the next time period is informed only by the previous one.

$$P(\Delta T_0, \Delta T_1, \dots, \Delta T_5) = P(\Delta T_0) * P(\Delta T_1 | \Delta T_0) * \dots * P(\Delta T_5 | \Delta T_4) \quad (4)$$

Combining Eqs. (3) and (4), we relate the joint density of the temperature change probabilities to the Bayesian model from Eqs. (1) and (2):

$$P(\Delta T_0 = \Delta X_0, \Delta T_1 = \Delta V_{1,i}, \dots, \Delta T_5 = \Delta V_{5,m})$$
$$= P(\Delta T_0 = \Delta X_0) * P(\Delta T_1 = \Delta V_{1,i} | \Delta T_0 = \Delta X_0) * \dots * P(\Delta T_5 = \Delta V_{5,m} | \Delta T_4 = \Delta V_{4,l})$$
$$= P(\mu_0 = \Delta X_0) * P(v_1 = \Delta V_{1,i}) * \dots * P(v(\Delta V_{4,l}) = \Delta V_{5,m}) \quad (5)$$
$$\forall i, j, k, l, m = 1, \dots, N$$

Next, we develop a joint distribution for the absolute mean temperatures in each time period, which correspond to the SDP temperature states $\mathbf{S}^T$. To do this, we assume $T_0 = X^* + \mu_0$, where $X^*$ is a constant reflecting the historical observed temperature in time $t-1$, and recognize that the absolute temperature in $t$ is the sum of all the relative changes between 0 and $t$ plus $T_0$. The joint density of the temperature states is therefore:

$$P(T_0 = a, T_1 = b, \dots, T_5 = f)$$
$$= P(\mu_0 = a - X^*) * P(v_1 = b - a) * \dots * P(v(\Delta V_{4,l}) = \Delta V_{5,m}) \quad (6)$$
$$\forall i, j, k, l, m = 1, \dots, N$$

where

$$a = X^* + \Delta X_0, \ b = X^* + \Delta X_0 + \Delta V_{1,i}, \dots,$$
$$f = X^* + \Delta X_0 + \Delta V_{1,i} + \Delta V_{2,j} + \Delta V_{3,k} + \Delta V_{4,l} + \Delta V_{5,m}$$
$$\text{s.t. } a, b, c, d, e, f \in \mathbf{S}^T$$

The SDP temperature transition probabilities consist of adjacent time period conditional probabilities, i.e. $P(T_t = w \mid T_{t-1} = v)$. We use Monte Carlo simulation to calculate them by sampling from the joint density in Eq. (6) as follows. First, sample from Eq. (6) to generate $M$ equally likely realizations of the joint density. Each realization forms a set, $\mathbf{Y}_i$, of the form:

$$\mathbf{Y}_i: \{T_0 = y_{0i}, T_1 = y_{1i}, T_2 = y_{2i}, T_3 = y_{3i}, T_4 = y_{4i}, T_5 = y_{5i}\} \forall i = 1, \dots, M$$

Second, let $R$ equal the number of sets $\mathbf{Y}_i$ out of the total of $M$ for which $T_t = w$ and $T_{t-1} = v$. Third, let $Q$ equal the number of sets $\mathbf{Y}_i$ out of the total of $M$ for which $T_{t-1} = v$. Then, the transition probabilities are:

$$P(T_t = w | T_{t-1} = v) = P(T_t = w, T_{t-1} = v)/P(T_{t-1} = v) = R/Q \quad (7)$$
$$\forall w, \ v \in \mathbf{S}^T$$

**Stochastic dynamic programming (SDP).** Stochastic dynamic programming is an optimization approach and control method that represents decision-making under uncertainty using multiple stages or time periods. The result is optimal policies, representing the best possible action as a function of the system state and time period. In our non-stationary formulation, it can also be understood as a form of closed-loop stochastic control, in which new information about the system feeds back into updated estimates for system state transitions over time. This is analogous to existing approaches in ecology, which have defined SDP transition probabilities with probability density functions that include the current system state as an input[45,46].

Optimal policies are derived by recursively solving the Bellman equation:

$$\mathbf{V}_t(\mathbf{s}_t) = \text{argmin}_{a \in \mathbf{A}} C(\mathbf{s}_t, a_t, t) + \gamma * \Sigma_{s \in \mathbf{S}} P(s_{t+1} | s_t, a_t) * \mathbf{V}_{t+1}(\mathbf{s}_{t+1}) \quad (8)$$

Where $t \in \{1 \dots 5\}$ is a 20-year time period ranging from 2001–2020 for $t=1$ to 2081–2100 for $t=5$. $\mathbf{S}$ is the state space, or set of possible state values, which includes: mean temperature $\mathbf{S}^T$ (which ranges from 25 to 33 °C at 0.05 °C increments) and mean precipitation $\mathbf{S}^P$ averaged over a 20-year period (which ranges from 66 to 97 mm month$^{-1}$ at 1 mm month$^{-1}$ increments) and available infrastructure $\mathbf{S}^Z$. $\mathbf{S}^Z = \{1,\dots,4\}$ which correspond, respectively, to a small infrastructure alternative, large infrastructure alternative, flexible unexpanded alternative, and flexible expanded alternative. The alternatives include a set of dams (planning scenarios A and B) or a set of desalination plants (planning scenario C). $\mathbf{s}_t \in \mathbf{S}$ is the system state at time $t$ comprised of $T_t$ (temperature state at time $t$), $P_t$

(precipitation state at time $t$), and $Z_t$ (infrastructure state at time $t$). $T_t$, $P_t$ and $Z_t$ are assumed independent. Therefore, the transition probabilities, P($s_{t+1}$ | $s_t$, $a_t$) * $V_{t+1}(s_{t+1})$, are estimated as three independent transition vectors: the transition vector for $T_t$ is described in Eqs. (4) and (5) and independent of $a_t$ and $P_t$ is analogous to $T_t$. $Z_t$ transitions are deterministic based on the current capacity and action to add capacity as described hereafter.

A is the action space, or set of possible actions. $a_t \in A$ is an action at time $t$, The action $a_t$ describes whether a static or flexible dam is chosen, and whether infrastructure capacity is expanded in later time periods. $A = \{0,...,4\}$ which correspond, respectively, to no change, adding a small static alternative, adding a large static alternative, adding a flexible alternative, and expanding the flexible alternative. The choices are constrained by time period and available infrastructure such that $a_{t=1} \in \{1,...,3\} \ \forall \ Z_t$; $a_t \in \{0,4\}$ when $Z_t = 3 \ \forall \ t = 2,...,5$; $a_t \in \{0\} \forall \ Z_t = \{1,2,4\}$, $t = 2,...,5$.

V is the optimal policy or choice of action. $\gamma$ is the discount rate. Costs C include the capital costs of infrastructure and damages if the infrastructure fails to meet reliability targets such that $C = I \ (s_t, a_t) + D * U(s_t, a_t)$, where $I$ is the cost of the infrastructure including capital costs (capex) and operating costs (opex). Desalination opex in planning scenario A is a function of the water produced in each time period. $D$ is unit cost of damages incurred for unmet water demand, set at 15$ m$^{-3}$ in our base case based on estimates of water productivity in Kenya from the World Bank[47]. $U$ is the volume of unmet demand as a function of the climate states, existing infrastructure, and any new infrastructure brought online in time t. $U = 0$ in $t = 1$, reflecting that $t = 1$ is a planning and construction period and performance is not measured until the beginning of the second 20-year time period.

**Stochastic weather generation.** Climate impacts on river runoff depend on changes in month-to-month variability in precipitation and temperature in addition to changes in the mean. We model these two changes separately. To develop monthly time-series of $T$ and $P$, we follow the $k$-nearest neighbors (kNN) approach as described in ref. [48] applied to GCM projections. This non-parametric statistical approach allows us to impose the mean $T$ and $P$ from the SDP while also capturing the standard deviation in monthly values and month-to-month autocorrelation projected by the GCMs. This approach was chosen for its simplicity, ease of implementation, and application in long-term water supply; future studies could use other non-parametric approaches such as the local polynomial regression method developed in ref. [49]. For each 20-year time period, we employ the kNN approach to generate 100 samples of 20-year long monthly time-series of $T$ and $P$. The resulting time series are then applied to the rainfall-runoff model presented below.

**Rainfall-runoff model.** Next, the synthetic $T$ and $P$ time series are input to a hydrological model to assess the impacts on runoff. We use CLIRUN II, the latest in a family of hydrological models developed to assess the impact of climate change on runoff[50-53]. CLIRUN II is a two-layer, conceptual, lumped-watershed rainfall-runoff model. It averages soil parameters over the watershed and models runoff at one gauge station at the mouth of the basin. It can be run on a monthly or daily time step. Using the kNN generated samples of $T$ and $P$, CLIRUN II generates a corresponding 100 samples of 20-year long monthly timeseries of runoff.

CLIRUN II is calibrated using 14 years of monthly streamflow data. Only one streamflow gauge, RGS 3MA03, is available in the Mwache basin[31]. However, it is directly upstream of the dam location, making it representative for this study. The same monthly temperature and precipitation data from CRU used in the Bayesian climate analysis is used to calibrate CLIRUN II for consistency. This temperature and precipitation data is different than the local data used in the previous World Bank study[21], leading to different calibration results but similar performance (historical MAR: 113 MCM year$^{-1}$; World Bank MAR: 133 MCM year$^{-1}$; our MAR: 103 MCM year$^{-1}$). Our analysis using CLIRUN II and the reservoir sizing model confirms that the 80 MCM dam meets the reliability targets in the current and expected future climate but does not meet reliability targets if the climate gets substantially warmer and drier. The 120 MCM dam meets reliability targets across all projected future climates.

**Infrastructure costs and operations.** Capex and opex estimates for the small and large dams were developed using the cost tool from the previous World Bank study[21]. For the flexible dam, the cost per m$^3$ of additional capacity added is assumed to be 50% greater than that of the original capacity. Capex and opex estimates for the RO desalination plants were developed using the Cost Estimator tool from DesalData[54].

The infrastructure operation model includes fixed dam operations (and desalination operations when necessary) that seek to meet the specified yield target while accounting for dead storage, net evaporation, and environmental flows. Unmet demand is measured for each of the 100 streamflow time series, and the average 20-year unmet demand is used to characterize $U$ in the SDP formulation in Eq. (8). We acknowledge that assuming reservoir operations that are fixed in time is a limitation given that adaptive reservoir operations would likely reduce the need for additional capacity; future work could optimize the reservoir operations to each climate state.

## Data availability

Historical climate data (CRU TS3. 10) is available here: https://crudata.uea.ac.uk/cru/data/hrg/. GCM projections are publicly available from the respective sources listed in Supplementary Table 1. Streamflow data is available in Supplementary Data 1.

## Code availability

Code is available from the corresponding author upon reasonable request.

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

## Acknowledgements

The authors are grateful for input and feedback from Dara Entekhabi, Olivier de Weck, James Wescoat, Afreen Siddiqi, and Adnan Alsaati. S.F. was supported by a Rasikbhai L. Meswani Fellowship for Water Solutions from the Abdul Latif Jameel Water and Food Systems Lab (J-WAFS) at MIT as well as a National Science Foundation Graduate Research Fellowship. M.L. acknowledges financial support from a Callahan-Dee Fellowship. Additional research funding was provided by the Center for Complex Engineering Systems at MIT and KACST.

## Author contributions

S.F. conceptualized the study. S.F., M.L., and K.S. designed the methodology. S.F. and M.L. performed the analysis. S.F. and M.L. wrote the manuscript. S.F., M.L., and K.S. edited the manuscript.

## Additional information

**Competing interests:** The authors declare no competing interests.

