## [Peer Review File · Nature Communications]

Reviewers' comments:

Reviewer #1 (Remarks to the Author):

This manuscript presents a useful framework for systematically comparing tradeoffs among flexible and static water infrastructure designs using a Bayesian model of learning to estimate future reduction in uncertainty about climate change. Such a framework is important because flexible design is increasingly recommended as a means to address climate change (see, for instance the new guidance from the State of California "Paying it Forward: The Path Toward Climate-Safe Infrastructure in California), but systematic approaches remain lacking for estimating what we may know in the future about climate change.

I recommend that this paper be published after the authors address some important issues as suggested below.

As my first general comment, I'd like to suggest that the authors might do a better job of situating their work in the literature. As the manuscript notes, the water sector has increasingly been addressing developing robust strategies that perform well over a wide range of plausible future scenarios. The manuscript is mistaken, however, in arguing that this previous work takes a "static" view of uncertainty. From the beginning, much of this work has considered learning about climate, generally in the form of a two or multi-period sequential decision framework in which improved information about climate change becomes available, often from unspecified sources, at specific times in the future. For one strand of such work see: Groves (2005), Groves and Lempert (2007), Groves et al. (2008), Lempert and Groves (2010), Groves et al. (2013), Groves et al. (2014). (Part of this manuscript's confusion regarding the treatment of uncertainty and learning may be its reliance on reference 11, which appears to have significantly mis-characterized the literature and in addition confused the concepts of robustness as a decision criterion with the concept of flexible vs. static as design strategies. More on this below.) In more recent work, Bloom (2015) updated the sequential decision analysis originally done by the Bureau of Reclamation (2012) using Bayesian learning from: 1) local stream flow observations and 2) unspecified advances in climate science that narrow the range of GCM projections. The current manuscript provides an important and welcome advance by incorporating into the decision analysis an explicit process of Bayesian learning on an ensemble of climate models. While there are examples of Bayesian learning from climate models in the sea level rise decision literature (Wong and Keller 2017), this paper would be the first that I know of in the water management literature.

As my second general comment, I'd encourage the authors to say more about their modeled process of learning about climate change and to the extent to which it represents the learning which may actually occur. I may have missed it, but I didn't see a clear discourse on whether the learning about temperature and precipitation referred to local, regional, or global patterns. This can be important, because for some locations, some GCMs get the local right and global wrong and visa versa. In addition, some models may do better than others on large scale climate patterns (e.g. ENSO) related to drought and other extreme events. I am not sure the extent to which that is important or relevant to the Mombasa location of the reservoir addressed in this manuscript, but in some locations such issues are important. This manuscript also appears to assume that some of the GCMs are structurally correct and others are not correct, and that this property of correctness is invariant over the century time scales considered in the analysis. That is, the analysis assumes that whatever bio-geophysical processes give one GCM more skill than another in 2020 will be the same as those most relevant to GCM skill in 2100 after a century of climate and other change. I'd find it helpful if the authors would address that assumption. Finally, I was surprised by the statement that the structural uncertainty among the GCMs dominates uncertainty in emissions past 2050. I didn't have time to look at the cite (#28) justifying this claim, but I find it surprising.

My more focused points include:

- The manuscript begins with cites to the robust decision making, decision making under deep uncertainty (DMDU) literature. But in the end this manuscript adopts a decision analytic framework seeking optimal strategies using well-characterized uncertainty. That is fine, and as the authors note, the analysis in this paper could be combined with vulnerability analyses/climate stress tests that explored the consequences of mis-characterizing the climate uncertainty. But the authors might make clearer what they have and have not done. In addition, the authors might, without too much extra work, conduct a vulnerability analyses/climate stress test by using the data in Figure 3 to calculate regret. For instance, using the data in Fig 3a, they might calculate the size of the error if one chooses a static dam design in a low T high P future or a flexible dam in a high T low P future.
- The authors might be more precise about the use of the word "robust." The word has many meanings in various literatures, but in the literature cited at the start of the manuscript, robustness is a decision criterion. Thus, it doesn't make sense to compare a "robust" dam to a "flexible" dam, any more than it makes sense to compare an "optimal" dam to a "flexible" dam. As the manuscript demonstrates, in some cases a static design is more robust than a flexible one, and in some cases the reverse is true. The manuscript would be clearer if its terminology focused on comparing static and flexible designs.
- The authors might clearly define the four strategies they consider in the main text. The clearest statement I found was in the supplement.

REFERENCES

- Bloom, E. (2015). Changing Midstream: Providing Decision Support for Adaptive Strategies Using Robust Decision Making.RGSD-348
- Bureau of Reclamation (2012). Colorado River Basin Water Supply and Demand Study: Study Report
- Groves, D. G., J. R. Fischbach, E. Bloom, D. Knopman and R. Keefe (2013). Adapting to a Changing Colorado River: Making Future Water Deliveries More Reliable Through Robust Management Strategies.RR242
Uncategorized References
- Groves, D. G. (2005). "New Methods for Identifying Robust Long-term Water Resources Management Strategies for California." Pardee RAND Graduate School PhD.
- Groves, D. G., E. W. Bloom, R. J. Lempert, J. R. Fischbach, J. Nevills and B. Goshi (2014). "Developing Key Indicators for Adaptive Water Planning." J. Water Resour. Plann. Manage.: 05014008-05014001 - 05014008-05014010.
- Groves, D. G., M. Davis, R. Wilkinson and R. Lempert (2008). "Planning for Climate Change in the Inland Empire: Southern California." Water Resources IMPACT July.
- Groves, D. G. and R. J. Lempert (2007). "A New Analytic Method for Finding Policy-Relevant Scenarios." Global Environmental Change 17: 73-85.
- Lempert, R. and D. G. Groves (2010). "Identifying and Evaluating Robust Adaptive Policy Responses to Climate Change for Water Management Agencies in the American West." Technological Forecasting and Social Change 77: 960-974.
- Wong, T. E. and K. Keller (2017). "Deep Uncertainty Surrounding Coastal Flood Risk Projections: A Case Study for New Orleans." AGU Publications, Earth's Future 2: 1015-1026.

Reviewer #4 (Remarks to the Author):

The authors propose a formal planning framework for evaluating the tradeoffs between flexible and traditional robust planning approaches. The same framework can also be used to assess the potential to learn about climate change over time, and evaluate flexible approaches. The procedure is exemplified on a reservoir planning problem in Mombasa, Kenya.

As the authors rightly point out in the introduction, flexible planning approaches play a critical role in climate change adaptation strategies, particularly in developing countries. To develop a formal approach for adaptive planning is therefore very welcome. I also agree with the authors that including a Bayesian learning step as an integral part of the framework is the most natural and satisfactory formal procedure to adopt. As the authors indicate in the abstract, their paper can be read in two ways: (i) as a way of deciding whether an adaptive planning approach is supposed to be taken in the first place, or (ii) as a way of evaluating different options within an adaptive approach at different points in time.

My comments here focus on the Bayesian aspect of the decision framework, and mainly concern perspective (i), i.e., apply to the situation in which a decision maker is faced with the choice between either opting for an flexible or a robust approach. This is a long-term decision, with a time horizon of about 50-100 years.

This decision is mainly based on the transition probabilities in the, so-called, SDP (Equation 1). My main concern is related to these transition probabilities. As the authors describe in the first section, at time zero t_0 , the data from climate model simulations are used as "virtual observations", observed at future times t_1 , t_2 , t_3 , and t_4 . In other words, in order to take a decision on whether to adopt a flexible or a robust planning approach, it is assumed that in the future we will actually observe the data which come from the climate model projections generated at time t_0 .

What is the effect of this assumption? As one can see in Figure 2 c-f, the median value of the sequentially updated probability distributions at time t_k are close to the median value of the posterior probability distribution of time t_0 (at time t_k). For example, in panel 2d, the median of the light blue probability distribution at time 2070 is close to the median of the dark blue probability distribution at time 2070. This is to be expected, since it is assumed that the data from the climate model projections performed at time t_0 will indeed be observed in the future. I think that this assumption is particularly problematic with regard to variables where the uncertainty range is large, like precipitation. The models do not agree on the sign of the change in precipitation in many regions, like in the example in the manuscript. The effect is that the model median is close to zero. But it is likely (say, 60% chance) that there will be either a consistent drying or a consistent wettening over time. In this case the actual future observations will consistently show either a drying or a wettening (but as a matter of fact we don't know at time t_0 which one).

The assumption that we will indeed observe the "virtual observations" generated by the models leads to sequentially updated posterior distributions which become narrower over time (which is correct), but the uncertainty in the future development of the median value of these distributions is not accounted for adequately.

I think that the effect of this is that the procedure actually tends to penalize the adaptive planning approach against the robust planning approach because the uncertainty is underestimated. In reality the uncertainty is larger than what the distributions in Figure 2 suggest. For example, the dark blue distribution in Figure 2d suggests that in 2070 we will be quite certain that the precipitation change in the year 2090 is essentially the median indicated by the climate projections

performed in the year 2010. And again, this is a general feature of the procedure. But in reality, in the year 2010, we are not sure whether it will rain more or less over Mombasa, and whether the median of the dark blue distribution will be positive or negative.

One can say: "but this is the best we have right now". I think this is not quite correct. The model does make implicit assumptions. The climate projections performed in the year 2010 indicate that the uncertainty about what will happen in the year 2090 is large. The sequential updating approach tends to underestimate the uncertainty by assuming that at time t_k we will actually observe what the climate models projected at time t_0 . In my opinion the procedure thus overemphasizes the information provided by the climate models, and in particular the compromise between the models as represented by the median of the projected "virtual observations". The resulting underestimation of uncertainty might disadvantage the adaptive planning option compared to the robust planning option.

Ultimately, when it comes to the decision whether to adopt an adaptive or a robust planning procedure, an approach involving the minimization of regret might still be preferable. This could still be framed based on a Bayesian statistical model, but acknowledging the full uncertainty for the year 2090 as projected by the models in the year 2010.

The proposed procedure is not problematic when applied step by step within an adaptive planning approach, with a time horizon of 20 years or so (perspective (ii) above). Then essentially the Bayesian analysis would just include one step, as described in Smith et. al. (2009), which is certainly a sound procedure.

Minor comments:

(1) Define GCM in line 37

(2) What emission scenario(s) was/were used for the climate projections?

(3) The selection of models, and the number of chosen ensemble members (SI, Table 1), will have a substantial impact on the result of the analysis. The selection seems a bit unusual. What were the considerations here?

(4) I am not completely sure I understand Figure 3. In Figure 3a, what is T and P exactly (at what point in time)? Is it T and P at time 2010, when the decision about adopting a flexible approach has to be taken?

(5) It might be necessary to put the work in a somewhat wider perspective. I can see that the amount of text is limited. But what is exactly the main novel aspect of the work? Is it mainly the combination of the SDP with a Bayesian analysis? For a review in another discipline see e.g. Yousefpour et al. (2012), A review of decision making approaches to handle uncertainty and risk in adaptive forest management under climate change, *Annals of Forest Science*, 69, 1-15.

Reviewer #5 (Remarks to the Author):

- What are the major claims of the paper?

The paper is a case study on using flexible adaptation hydraulic structures for water resources projects. The study focuses on a region of Kenya where water availability is a challenge, and where a dam was proposed to be built. Based on a previous study indicating the costs of building a small or large dam, the authors propose to use a stochastic dynamic programming method to sequentially optimize and determine if the dams should be raised or not depending on climate evolution, remaining dam life and cost to upgrade. The authors state that this method could be used by practitioners to optimize dam costs and better manage climate change risk to water infrastructure.

- Are the claims novel? If not, please identify the major papers that compromise novelty

Although the implementation using SDP for this purpose is novel to me, the idea of rethinking structural analysis and also operational adaptation (which is not dealt with in this paper) of dams is not new:

Ehsani, N., Vörösmarty, C.J., Fekete, B.M. and Stakhiv, E.Z., 2017. Reservoir operations under climate change: storage capacity options to mitigate risk. *Journal of Hydrology*, 555, pp.435-446.

Watts, R.J., Richter, B.D., Opperman, J.J. and Bowmer, K.H., 2011. Dam reoperation in an era of climate change. *Marine and Freshwater Research*, 62(3), pp.321-327.

Cherry, J.E., Knapp, C., Trainor, S., Ray, A.J., Tedesche, M. and Walker, S., 2017. Planning for climate change impacts on hydropower in the Far North. *Hydrology and Earth System Sciences*, 21(1), pp.133-151.

Cervigni, R., Liden, R., Neumann, J.E., Strzepek, K.M. (Eds.), 2015. *Enhancing the Climate Resilience of Africa's Infrastructure: The Power and Water Sectors*. The World Bank.

- Will the paper be of interest to others in the field?

I think there will be interest from a small group of researchers and practitioners, but they are unlikely to try and find this information in a *Nature Communications Journal*. I believe this work is best suited for a more specific, technical journal such as *Journal of hydrologic engineering* or *Water Resources Management*.

- Will the paper influence thinking in the field?

I believe that it is a nice addition (in implementation and methodology) for specialists in this field, but is unlikely to influence the general mindset. There are too many hypotheses related to this case study to justify the generalization to other sites and uses (see comments below on the weather generator component). Furthermore, it is likely that adapting operation rules according to the changes in climate will help mitigate a certain amount of the risk.

- Are the claims convincing? If not, what further evidence is needed?

The general premise is good and convincing, but some details make the results less conclusive to me, and especially not generalizable. For example, the methodology depends on a weather generator to simulate trajectories. The block bootstrapping method used here does not preserve autocorrelations on longer timescales (i.e yearly) due to natural variability, thus averaging-out some of the variability that is expected to occur on longer timescales. Also, the monthly timestep is adequate for this volumetric study, but in no case could it be used for most systems which are managed on a daily or weekly time step. At these timescales, other interactions must be taken into account which this study fails to do. For example, extreme precipitation might drive the need for a larger reservoir, but if all we have is an increase of 10% in monthly precipitation, then this extreme event will not be taken into account. These are cases where management rules can be updated to adapt to climate change. I understand that the authors have a specific case in mind, and that the adaptations could still be used to further reduce dam size during initial sizing of the dams, but as it stands it is difficult for me to suggest that this paper is of broad enough interest for publication in a *Nature* journal.

- Are there other experiments that would strengthen the paper further? How much would they improve it, and how difficult are they likely to be?

Some experiments that could be done would be to use an SDP to re-optimize the management rules in time, and condition their work on that. Also, smaller reservoirs with more reactive times would be useful for the broader community. These additions would probably be difficult enough to put in place, so this is why I recommend the authors submit their work to a journal where their (very nice) incremental addition can be better positioned.

- Are the claims appropriately discussed in the context of previous literature?

Yes, the authors did a good job in their literature review and placed their work in context.

- If the manuscript is unacceptable in its present form, does the study seem sufficiently promising that the authors should be encouraged to consider a resubmission in the future?

As per my points above, I think the subject is good, the authors did a good job for their case study, but that it is simply too narrow of a research field for this journal.

Reviewer #1 (Remarks to the Author):

This manuscript presents a useful framework for systematically comparing tradeoffs among flexible and static water infrastructure designs using a Bayesian model of learning to estimate future reduction in uncertainty about climate change. Such a framework is important because flexible design is increasingly recommended as a means to address climate change (see, for instance the new guidance from the State of California “Paying it Forward: The Path Toward Climate-Safe Infrastructure in California), but systematic approaches remain lacking for estimating what we may know in the future about climate change.

Thank you. We agree this is the key contribution of the paper.

I recommend that this paper be published after the authors address some important issues as suggested below.

As my first general comment, I'd like to suggest that the authors might do a better job of situating their work in the literature. As the manuscript notes, the water sector has increasingly been addressing developing robust strategies that perform well over a wide range of plausible future scenarios. The manuscript is mistaken, however, in arguing that this previous work takes a “static” view of uncertainty. From the beginning, much of this work has considered learning about climate, generally in the form of a two or multi-period sequential decision framework in which improved information about climate change becomes available, often from unspecified sources, at specific times in the future. For one strand of such work see: Groves (2005), Groves and Lempert (2007), Groves et al. (2008), Lempert and Groves (2010), Groves et al. (2013), Groves et al. (2014). (Part of this manuscript's confusion regarding the treatment of uncertainty and learning may be its reliance on reference 11, which appears to have significantly mischaracterized the literature and in addition confused the concepts of robustness as a decision criterion with the concept of flexible vs. static as design strategies. More on this below.

In more recent work, Bloom (2015) updated the sequential decision analysis originally done by the Bureau of Reclamation (2012) using Bayesian learning from: 1) local stream flow observations and 2) unspecified advances in climate science that narrow the range of GCM projections. The current manuscript provides an important and welcome advance by incorporating into the decision analysis an explicit process of Bayesian learning on an ensemble of climate models. While there are examples of Bayesian learning from climate models in the sea level rise decision literature (Wong and Keller 2017), this paper would be the first that I know of in the water management literature.

We appreciate your comments and agree that the description of previous work, especially RDM, can be improved. We acknowledge that RDM has been used to evaluate adaptive infrastructure approaches, and that Walker et al. (2013)'s classification of different methods in the field is controversial. We agree also that our use of the terms dynamic, static, and robust was imprecise, and that the main

contribution is the integration of Bayesian learning into a multi-stage decision process as a way to better assess the tradeoffs of flexible vs. static approaches upfront. We appreciate the suggested references.

We have addressed these concerns in the following ways:

- We have aligned terminology throughout the paper. We now use static or large to describe the larger dam, rather than robust. The term dynamic is now used to refer to planning processes with multiple stages where actions can be changed in the future (rather than to incorporating a learning feedback).
- The introduction section has been rewritten to:
 - clarify the terminology
 - focus the literature review on adaptive approaches and include applications of RDM used for adaptive planning (Groves and Lempert 2007, Groves et al 2008, Lempert and Groves 2010, Groves et al 2015, Bloom 2015 PhD dissertation)
 - Emphasize the integration of Bayesian learning into the planning model to assess irreversible infrastructure decisions as the main contribution
- The abstract has also been updated to reflect clarified terminology and emphasis also

Regarding our contribution of integrating Bayesian learning into an adaptive decision framework, we would like to add one note: While we agree that Wong and Keller 2017 is an excellent example of Bayesian probabilities used for sea level rise decision-making, we see our contribution as greater than merely extending this to water management. Our approach connects the Bayesian probabilities to an iterative learning process where the Bayesian probabilities are updated in each stage of an adaptive planning process, while Wong and Keller 2017 develop one-time Bayesian probabilities that are not updated or connected to adaptive infrastructure options. Our approach could therefore have impactful extensions to sea-level rise planning and many other infrastructure-focused climate change adaptation domains as well.

As my second general comment, I'd encourage the authors to say more about their modeled process of learning about climate change and to the extent to which it represents the learning which may actually occur. I may have missed it, but I didn't see a clear discourse on whether the learning about temperature and precipitation referred to local, regional, or global patterns. This can be important, because for some locations, some GCMs get the local right and global wrong and visa versa. In addition, some models may do better than others on large scale climate patterns (e.g. ENSO) related to drought and other extreme events. I am not sure the extent to which that is important or relevant to the Mombasa location of the reservoir addressed in this manuscript, but in some locations such issues are important.

We agree that the learning will differ depending on whether we are learning about regional vs global patterns or a GCM's ability to reproduce teleconnections. Based on previous work, it appears that while ENSO teleconnections exert controls over the

annual rainfall variability both now and in a future warmer climate (e.g. Funk et al., 2014; Lickley et al. 2018), long-term trends in rainfall in East Africa are likely influenced by warming in the Indian Ocean (Funk et al., 2008). Evaluating models based on their representation of ENSO teleconnections could provide some useful insight into potential early warning signals, though it is difficult to say if that would necessarily improve learning. For instance, if a model is good at reproducing ENSO teleconnections in Mombasa, but poor at reproducing annual rainfall trends in the region, it is unclear that it would provide useful insight into the planning processes employed here.

Here, we frame learning around regional climate change because of the geographic scale of the proposed project. That is to say that local, rather than global, climate change will be of paramount concern to water resource planners in the Mwache catchment. We have added some clarifying language in the text to make this more specific and explain the rationale:

Clarification in methods section (Lines 571-573):

These regional temperature and precipitation GCM outputs, rather than global outputs, provide the basis for model weighting in the Bayesian analysis.

And (Lines 553-559)

The evaluation of GCMs' performance in reproducing climate observations will depend on time scale, region, and variable of interest (Johnson et al., 2011; Flato et al., 2013). Because our ultimate goal is to update our learning of regional climate in the Mwache catchment with respect to multi-decadal trends in precipitation and temperature, we choose to weight GCMs based on their performance in reproducing multi-decadal trends of precipitation and temperature averaged over the catchment area.

Additionally, we have added "regional" in several places throughout the paper when we describe historical and future temperature and precipitation change, including on line 72 when we define the T and P shorthand we use throughout the paper.

Flato, G., J. Marotzke, B. Abiodun, P. Braconnot, S.C. Chou, W. Collins, P. Cox, F. Driouech, S. Emori, V. Eyring, C. Forest, P. Gleckler, E. Guilyardi, C. Jakob, V. Kattsov, C. Reason and M. Rummukainen, 2013: Evaluation of Climate Models. In: Climate Change 2013: The Physical Science Basis. Contribution of Working Group I to the Fifth Assessment Report of the Intergovernmental Panel on Climate Change [Stocker, T.F., D. Qin, G.-K. Plattner, M. Tignor, S.K. Allen, J. Boschung, A. Nauels, Y. Xia, V. Bex and P.M. Midgley (eds.)]. Cambridge University Press, Cambridge, United Kingdom and New York, NY, USA.

Funk, C., Hoell, A., Shukla, S., Blade, I., Liebmann, B., Roberts, J. B., ... & Husak, G. (2014). Predicting East African spring droughts using Pacific and Indian Ocean sea surface temperature indices. *Hydrology and Earth System Sciences*, 18(12), 4965-4978.

Funk, C., Dettinger, M. D., Michaelsen, J. C., Verdin, J. P., Brown, M. E., Barlow, M., & Hoell, A. (2008). Warming of the Indian Ocean threatens eastern and southern African food security but could be mitigated by agricultural development. *Proceedings of the national academy of sciences*, 105(32), 11081-11086.

Johnson, F., Westra, S., Sharma, A., & Pitman, A. J. (2011). An assessment of GCM skill in simulating persistence across multiple time scales. *Journal of Climate*, 24(14), 3609-3623.

Lickley, M., & Solomon, S. (2018). On the relative influences of different ocean basin sea surface temperature anomalies on southern African rainfall in 20th and 21st century general circulation model simulations. *International Journal of Climatology*, 38(13), 5003-5009.

To your broader point about to what extent this GCM-based Bayesian learning model realistically reflects planners' potential learning in an iterative, adaptive planning process, we acknowledge that this is an imperfect measure but argue that it is still useful for weighing initial tradeoffs about irreversible infrastructure decisions. All planning methods that use GCM projections face limitations. The key is in how they are used and interpreted. We here are not suggesting that the probability distributions estimated are completely precise, unbiased, consistent estimators of future uncertainty. Rather, they allow for a reasonable quantification of opportunities to learn in the future, enabling better informed comparison of static vs. flexible approaches upfront, which is necessary for irreversible infrastructure investments. We have updated the first paragraph in the discussion section to acknowledge this more clearly:

“While all approaches that use GCM ensembles face limitations, this approach provides a reasonable quantitative estimate of future learning that enables better-informed assessment of tradeoffs between planning approaches. This allows us to evaluate the effectiveness of flexible planning, which relies on learning processes that remain unquantified in previous methods, rather than assuming a priori that flexibility is a worthwhile planning goal. This is especially important for infrastructure planning where planners must prepare in advance to take a flexible approach due to the large, irreversible nature of infrastructure investments.”

This manuscript also appears to assume that some of the GCMs are structurally correct and others are not correct, and that this property of correctness is invariant over the

century time scales considered in the analysis. That is, the analysis assumes that whatever bio-geophysical processes give one GCM more skill than another in 2020 will be the same as those most relevant to GCM skill in 2100 after a century of climate and other change. I'd find it helpful if the authors would address that assumption.

The reviewer raises an important point that the analysis assumes that model skill in reproducing historical processes is related to model skill in projecting future outcomes. This could be misleading, especially on long time scales, and as the reviewer suggests we lack definitive evidence that a model's skill in hindcasting actual observations suggests skill in producing accurate forecasts.

However, in Smith et al., (2009), whose model has been adopted here, provide a cross-validation for the method. This cross-validation assumes that each of the GCMs simulations can be treated as a random draw of possible climate outcomes. They then select one random of GCM to be removed from the ensemble and treated as an observed climate trajectory. It is then validated that the Bayesian modeling approach is able to correctly reproduce realistic projections and uncertainties of the removed GCM. They test this cross-validation method for each GCM separately. We have noted this in the text on line 604.

Further, in our adoption of the Smith et al., (2009) model we limit forecasting to changes in 20-year increments instead of the longer timeframe used in Smith et al., (2009). Therefore, we only assume that the modeled rate of change of the climate system in 2020, for example, would provide insight into the modeled rate of change in 2040, and not in time periods beyond 2040. This therefore does allow for varying weights on GCM reliability over time.

Finally, I was surprised by the statement that the structural uncertainty among the GCMs dominates uncertainty in emissions past 2050. I didn't have time to look at the cite (#28) justifying this claim, but I find it surprising.

Hawkins and Sutton (2010) have provided a helpful framework for thinking about climate uncertainty. To summarize, over short time frames, the climate signal can be masked by natural fluctuations of the climate system. Over longer timer periods, the climate signal will emerge from the noise. The extent to which the climate signal emerges is a function of the model's climate sensitivity and hence model structure. We've added a clarifying statement in the methods section (Line 507):

Hawkins and Sutton, (2010) show that the uncertainty in climate projections due to natural variability remains relatively constant throughout the 21st century, but that as the climate signal emerges from the noise, the uncertainty in projections is dominated by the GCMs' climate sensitivity, and hence structure.

My more focused points include:

- The manuscript begins with cites to the robust decision making, decision making under deep uncertainty (DMDU) literature. But in the end this manuscript adopts a decision analytic framework seeking optimal strategies using well-characterized uncertainty. That is fine, and as the authors note, the analysis in this paper could be combined with vulnerability analyses/climate stress tests that explored the consequences of mis-characterizing the climate uncertainty. But the authors might make clearer what they have and have not done. In addition, the authors might, without too much extra work, conduct a vulnerability analyses/climate stress test by using the data in Figure 3 to calculate regret. For instance, using the data in Fig 3a, they might calculate the size of the error if one chooses a static dam design in a low T high P future or a flexible dam in a high T low P future.

We have included a simple regret analysis at the end of the results section. We have additionally clarified that the other parts of analysis were top-down in that they used probabilistic forecasts of GCMs and that the regret analysis is an example of combining with bottom-up analysis. See Figure 6 and accompanying the additional text below on lines 248-265:

“Finally, while the previous analysis has relied on a "top-down" analysis that uses GCM projections to develop probabilistic forecast, Figure 6 presents an illustrative "bottom-up" analysis that demonstrates the average cost and regret of each of the three dam alternatives in planning scenario A under different end-of-century climates without relying on probabilistic forecasts. Regret is defined as the difference between the cost of the chosen infrastructure alternative and the best possible infrastructure alternative. Three illustrative climates are chosen to demonstrate the tradeoffs across alternatives: a dry climate of 68 mm/month, a moderate climate of 78 mm/month, and a wet climate of 88 mm/month. Differences in T are not considered because its impact on water shortages is limited. The small dam without expansion has the highest maximum regret of any alternative of M\$77, incurred in the dry climate. The large dam incurs positive regret in both the moderate and wet climates, with the latter incurring M\$19 of regret. The flexible dam has the lowest maximum regret, with a modest M\$4 of regret in the dry climate. This bottom up approach also highlights the ability of the flexible dam design and expansion strategy to mitigate risk in a range of different potential future climates.”

The authors might be more precise about the use of the word “robust.” The word has many meanings in various literatures, but in the literature cited at the start of the manuscript, robustness is a decision criterion. Thus, it doesn’t make sense to compare a “robust” dam to a “flexible” dam, any more than it makes sense to compare an “optimal” dam to a “flexible” dam. As the manuscript demonstrates, in some cases a static design is more robust than a flexible one, and in some cases the reverse is true. The manuscript would be clearer if its terminology focused on comparing static and flexible designs.

We have clarified this as noted in our earlier response to your 1st general comment and adopted flexible vs. static as our terminology.

- The authors might clearly define the four strategies they consider in the main text. The clearest statement I found was in the supplement.

The description of the planning scenarios has now been moved from the methods section after the references to the results section on page 119-142.

REFERENCES

Bloom, E. (2015). Changing Midstream: Providing Decision Support for Adaptive Strategies Using Robust Decision Making.RGSD-348

Bureau of Reclamation (2012). Colorado River Basin Water Supply and Demand Study: Study Report

Groves, D. G., J. R. Fischbach, E. Bloom, D. Knopman and R. Keefe (2013). Adapting to a Changing Colorado River: Making Future Water Deliveries More Reliable Through Robust Management Strategies.RR242

Uncategorized References

Groves, D. G. (2005). "New Methods for Identifying Robust Long-term Water Resources Management Strategies for California." Pardee RAND Graduate School PhD.

Groves, D. G., E. W. Bloom, R. J. Lempert, J. R. Fischbach, J. Nevills and B. Goshi (2014). "Developing Key Indicators for Adaptive Water Planning." J. Water Resour. Plann. Manage.: 05014008-05014001 - 05014008-05014010.

Groves, D. G., M. Davis, R. Wilkinson and R. Lempert (2008). "Planning for Climate Change in the Inland Empire: Southern California." Water Resources IMPACT July.

Groves, D. G. and R. J. Lempert (2007). "A New Analytic Method for Finding Policy-Relevant Scenarios." Global Environmental Change 17: 73-85.

Lempert, R. and D. G. Groves (2010). "Identifying and Evaluating Robust Adaptive Policy Responses to Climate Change for Water Management Agencies in the American West." Technological Forecasting and Social Change 77: 960-974.

Wong, T. E. and K. Keller (2017). "Deep Uncertainty Surrounding Coastal Flood Risk Projections: A Case Study for New Orleans." AGU Publications, Earth's Future 2: 1015-1026.

Reviewer #4 (Remarks to the Author):

The authors propose a formal planning framework for evaluating the tradeoffs between flexible and traditional robust planning approaches. The same framework can also be used to assess the potential to learn about climate change over time, and evaluate flexible approaches. The procedure is exemplified on a reservoir planning problem in Mombasa, Kenya.

As the authors rightly point out in the introduction, flexible planning approaches play a critical role in climate change adaptation strategies, particularly in developing countries. To develop a formal approach for adaptive planning is therefore very welcome. I also agree with the authors that including a Bayesian learning step as an integral part of the framework is the most natural and satisfactory formal procedure to adopt. As the authors indicate in the abstract, their paper can be read in two ways: (i) as a way of deciding whether an adaptive planning approach is supposed to be taken in the first place, or (ii) as a way of evaluating different options within an adaptive approach at different points in time.

Thank you. We agree this is the key contribution of the paper.

My comments here focus on the Bayesian aspect of the decision framework, and mainly concern perspective (i), i.e., apply to the situation in which a decision maker is faced with the choice between either opting for an flexible or a robust approach. This is a long-term decision, with a time horizon of about 50-100 years.

This decision is mainly based on the transition probabilities in the, so-called, SDP (Equation 1). My main concern is related to these transition probabilities. As the authors describe in the first section, at time zero t_0 , the data from climate model simulations are used as "virtual observations", observed at future times t_1 , t_2 , t_3 , and t_4 . In other words, in order to take a decision on whether to adopt a flexible or a robust planning approach, it is assumed that in the future we will actually observe the data which come from the climate model projections generated at time t_0 .

What is the effect of this assumption? As one can see in Figure 2 c-f, the median value of the sequentially updated probability distributions at time t_k are close to the median value of the posterior probability distribution of time t_0 (at time t_k). For example, in panel 2d, the median of the light blue probability distribution at time 2070 is close to the median of the dark blue probability distribution at time 2070. This is to be expected, since it is assumed that the data from the climate model projections performed at time t_0 will indeed be observed in the future.

I think that this assumption is particularly problematic with regard to variables where the uncertainty range is large, like precipitation. The models do not agree on the sign of the change in precipitation in many regions, like in the example in the manuscript. The effect is that the model median is close to zero. But it is likely (say, 60% chance) that there will be either a consistent drying or a consistent wetting over time. In this case the actual future observations will consistently show either a drying or a wetting (but

as a matter of fact we don't know at time t_0 which one).

The assumption that we will indeed observe the "virtual observations" generated by the models leads to sequentially updated posterior distributions which become narrower over time (which is correct), but the uncertainty in the future development of the median value of these distributions is not accounted for adequately.

I think that the effect of this is that the procedure actually tends to penalize the adaptive planning approach against the robust planning approach because the uncertainty is underestimated. In reality the uncertainty is larger than what the distributions in Figure 2 suggest. For example, the dark blue distribution in Figure 2d suggests that in 2070 we will be quite certain that the precipitation change in the year 2090 is essentially the median indicated by the climate projections performed in the year 2010. And again, this is a general feature of the procedure. But in reality, in the year 2010, we are not sure whether it will rain more or less over Mombasa, and whether the median of the dark blue distribution will be positive or negative.

One can say: "but this is the best we have right now". I think this is not quite correct. The model does make implicit assumptions. The climate projections performed in the year 2010 indicate that the uncertainty about what will happen in the year 2090 is large. The sequential updating approach tends to underestimate the uncertainty by assuming that at time t_k we will actually observe what the climate models projected at time t_0 . In my opinion the procedure thus overemphasizes the information provided by the climate models, and in particular the compromise between the models as represented by the median of the projected "virtual observations". The resulting underestimation of uncertainty might disadvantage the adaptive planning option compared to the robust planning option.

Ultimately, when it comes to the decision whether to adopt an adaptive or a robust planning procedure, an approach involving the minimization of regret might still be preferable. This could still be framed based on a Bayesian statistical model, but acknowledging the full uncertainty for the year 2090 as projected by the models in the year 2010.

The proposed procedure is not problematic when applied step by step within an adaptive planning approach, with a time horizon of 20 years or so (perspective (ii) above). Then essentially the Bayesian analysis would just include one step, as described in Smith et. al. (2009), which is certainly a sound procedure.

Thank you for your detailed review of the Bayesian method used to characterize the transition probabilities. We think that your concern is actually a misunderstanding due to a lack of clarity in our presentation of the methods and results.

If we interpret your comments correctly, you have understood the procedure as the following: the starting observation in $t=0$ is used to develop a Bayesian uncertain

projection for precipitation in $t=1$. The median of this uncertain projection is assumed to be the virtual observation in $t=1$, which is then used to develop a Bayesian uncertainty distribution for $t=2$. The median of the new distribution for $t=2$ is assumed to be the virtual observation for $t=2$ and used to develop a new distribution for $t=3$. Etc., etc.

In fact, we do not assume that the median from the estimated distribution becomes the virtual observation in the next time period. Rather, we have many different possible virtual observations in each time period, and we estimate unique probability distributions for each of them. The virtual observations for precipitation change, for example, range from -30% to +30% per 20-year time period discretized at 2%, for a total of 31 unique virtual precipitation observations. Each of these 31 virtual observations in each of 5 time periods generates a unique probabilistic projection for precipitation change in the following time period (155 in total).

This means that we do not assume a priori that precipitation is getting wetter, drier, or staying close to the median, or even that the virtual observations have anything to do with the projections at $t=0$; we allow for all possibilities. Moreover, the power of using a stochastic optimization formulation is that it allows us to explicitly develop optimal policies under the uncertainty that a wide range of future virtual observations may occur.

We suspect that Figure 2 of the original draft has played a large role in this misunderstanding. Because it is difficult to visualize so many unique probability distributions, we chose to visualize one sample time series of virtual observations to demonstrate the process used. The time series we randomly chose happened to have very modest increases in precipitation, which we now see may lead the reader to assume that the virtual observations are equivalent to the previous median. The black dots are the virtual observations, chosen at random; no medians are illustrated on the figure. We agree that assuming only one path of observations would be highly problematic, and also that it would tend to underestimate the adaptive approach. We apologize for the confusion.

We have implemented a number of changes to clarify:

Figure changes:

- Figure 1 has been updated to illustrate the process more clearly. The bottom left panel illustrates the discretization of virtual observations relative to the GCM projections; each box is a virtual observation. The bottom right illustrates two sample virtual observations, both in the same time period, one wet and one dry. We hope this more clearly illustrates that there are many different virtual observations considered in each time period.
- The previous figure 2 (now figure 3) has been updated to include two sample time series, one dry and one wet.

Text changes:

- Lines 70-77 in the introduction paragraph that introduces the approach describes the virtual observations more clearly:
“First, we use GCM projections forced by a high emissions scenario (Representative Concentration Pathway 8.5) to develop a wide range of

possible future mean regional temperature (\$T\$) and precipitation (\$P\$) over a planning horizon. We finely discretize mean T and P within that range. This develops a comprehensive set of "virtual climate observations" of mean T and P that reflect many possible future regional climates, some of which are drier and some of which are wetter."

- The results section on lines 155-181 describing the updating process illustrated in Figure 3 (previously figure 2) has been updated to both describe two alternative scenarios and also give more detail on the range of virtual observations considered:

"While Figure 2 presents Bayesian CIs based on historical observations, the SDP transition probabilities require Bayesian uncertainty estimates that reflect what will have been learned for many possible virtual future observations. We assume that precipitation change will range between -30% and +30% by end of century; we discretize this range at 2% for a total of 31 unique virtual precipitation change observations. We apply the Bayesian uncertainty analysis to each of these 31 virtual precipitation change observations in each time period. For example, Two sample time series of virtual T and P observations and their corresponding updated uncertainty estimates are shown in Figure 3. An example of strongly increasing P is shown at top; an example of modestly decreasing P is at bottom. For each virtual observation, we simulate 10,000 virtual climate time series from the current observation to the end of the planning period and construct a 90% CI, shown by the shaded regions. This process is repeated for each time step, with darker colors in the plot corresponding to the CIs developed from virtual observations sampled later in the planning period. The darker CIs therefore reflect uncertainty estimates updated with information farther into the future. The sample of virtual observations showing strong increases in P (Figure 3 a-d), leads to high certainty by the end of the century that negligible water shortages will be incurred, assuming the small 80 MCM of dam capacity. The alternate sample of virtual observations showing modest decreases in P demonstrates a reduction in uncertainty in both P and MAR. Expected water shortages increase substantially as more observations are collected, and the uncertainty increase as well due to non-linear relationships between MAR and shortages."
- The description of the SDP results on lines 182-187 has been updated to more clearly articulate the connection of the Bayesian updating process to the SDP formulation.

"While two sample time series of observations are illustrated in Figure 2, the SDP optimal strategy accounts for a wide range of possible future observations and what would be learned if they were to be observed. This is achieved through the multistage stochastic optimization formulation, which allows for uncertain, rather than deterministic, transitions to new climate states in each period."
- We added a paragraph in the methods section on lines 535-552:

“Finally, we apply the model to 1) multiple pairs of time windows and 2) many virtual observations of historical change in T and percentage change in P . Smith et al. (2009) assumed two periods: a historical climate (1961-1990) and a future climate (2071-2100). We also use a historical and future climate in each estimation of the Bayesian model; however, we define 6 time periods using pairs of adjacent 20-year windows and calculate the change in T and percentage change in P between adjacent windows. This gives a total of 5 pairs of historical and future adjacent windows within 1960-2099. In each pair of adjacent windows, the "historical" window corresponds to the current time period in the SDP and the "future" window corresponds to the next 20-year period; this is necessary for the 1-stage transition probabilities needed in the SDP. The 20-year time interval was chosen so that interannual variability was not driving the trend in precipitation and temperature across time periods. Smith et al. (2009) used historical observations of climate data X_0 in Equation 1); we repeat the analysis many times using unique virtual climate observations. Virtual temperature change observations range from 0 to 1.5 degrees C using discrete steps of 0.05 degrees C. Virtual observations of percentage change in precipitation range from -30% to 30% using discrete steps of 2%.”

- We have added a paragraph on lines 652-668 of the methods section which describes more precisely how the uncertainty estimates from the Bayesian statistical model are used to characterize the transition probabilities in the SDP.

“Note that the Bayesian statistical model develops uncertainty distributions based on relative change from one period to the next. For example, we estimate the probability distribution of change in T from 2030 to 2050 given that we observed 0.5 degrees C of change from 2010 to 2030. The SDP, however, uses the absolute (rather than relative) values of T and P as these are needed to assess the performance of infrastructure across different climate states. We convert the distributions of relative change to absolute change using Monte Carlo simulation to estimate the conditional probabilities as follows: starting from an initial value of absolute temperature, we sample from the distributions of relative change from one time period to the next to develop simulated time series of absolute temperature. Then, for an absolute temperature value X in a time period t , we select only the set of time series S in which $T_t = X$, and use the distribution P of values of T_{t+1} in S to characterize the transition probability $p(T_{t+1} | T_t = X)$. This procedure estimates $P = p(T_{t+1} = Y \cap T_t = X) / p(T_t = X)$, which is equivalent to $p(T_{t+1} = Y | T_t = X)$ by the definition of conditional probability. An analogous procedure is implemented for P .”

Minor comments:

(1) Define GCM in line 37

This has been modified to define GCMs (Page 2 line 39):

“General Circulation Models (GCMs) (i.e. climate models)”

(2) What emission scenario(s) was/were used for the climate projections?

All the climate projections used were forced by RCP 8.5. We chose a single emissions scenario because the Bayesian modeling approach by Smith et al. 2009 was designed to represent structural uncertainty, not emissions uncertainty. Future work could extend the approach to other emissions scenarios. This has been clarified in the methods section (lines 566-567):

All models are forced by the RCP 8.5 scenario, which is the high emissions scenario from the IPCC AR5.

(3) The selection of models, and the number of chosen ensemble members (SI, Table 1), will have a substantial impact on the result of the analysis. The selection seems a bit unusual. What were the considerations here?

Our choice of 21 GCM simulations was based on readily available GCM simulations as well a compromise between computational efficiency and model diversity. It is also in the range of the number of GCM runs commonly used for this type of analysis (see for ex. Knutti et al., 2010). The Smith et al., (2009) paper, whose Bayesian methods we follow, use nine GCMs in their analysis. Here, we chose to use a wider range of available models to increase the potential range of model projections, however, given the computational requirements of processing GCMs and given the diminishing marginal returns with regards to the range of future climate outcomes for each additional GCM, we chose not to process more than 21 models. We clarified our rationale in the Methods section of the main text (Lines 561-567):

The 21 GCM simulations come from 10 different institutions and 15 different GCMs, with three GCMs providing more than one simulation. Models were selected based on the most readily available models at the time of the analysis, with 21 being in line with previous studies, providing a reasonable balance between computational limits and model diversity (Knutti et al., 2010). All models are forced by the RCP 8.5 scenario, which is the high emissions scenario from the IPCC AR5.

References:

Knutti, R., Furrer, R., Tebaldi, C., Cermak, J., & Meehl, G. A. (2010). Challenges in combining projections from multiple climate models. *Journal of Climate*, 23(10), 2739-2758.

Smith, R. L., Tebaldi, C., Nychka, D., & Mearns, L. O. (2009). Bayesian modeling of uncertainty in ensembles of climate models. *Journal of the American Statistical Association*, 104(485), 97-116.

(4) I am not completely sure I understand Figure 3. In Figure 3a, what is T and P exactly (at what point in time)? Is it T and P at time 2010, when the decision about adopting a flexible approach has to be taken?

Yes, that is the correct interpretation. This has been clarified both in the figure caption (now Figure 4) and on lines 187-189.

“In the first time period, shown in Figure 4 (a), the SDP develops a threshold as a function of T and P during the 2001-2020 time period when the initial infrastructure decision is made.”

(5) It might be necessary to put the work in a somewhat wider perspective. I can see that the amount of text is limited. But what is exactly the main novel aspect of the work? Is it mainly the combination of the SDP with a Bayesian analysis? For a review in another discipline see e.g. Yousefpour et al. (2012), A review of decision making approaches to handle uncertainty and risk in adaptive forest management under climate change, *Annals of Forest Science*, 69, 1-15.

Yes, the main contribution is using the Bayesian analysis to characterize the stochastic uncertainty in the SDP. The very nice review paper you suggested actually highlights this as a challenge in forest management as well (From page 8, emphasis added):

“Common to all of these problems and decision approaches is the ability of the researcher to validly describe the stochastic process for the uncertain variables.... [These] approaches have only rarely been used to climate change and uncertainty about the effect on, e.g. growth patterns of species under climate change. Thus, the change is non-stationary in a way that does not offer the researcher a firm fundament for, e.g. setting up a stationary state transition matrix, or even a time bounded non-stationary such. **Assessment of transition probabilities will change as new information about the transition and its implications arise.** “

This is precisely the challenge that our approach addresses! We use the Bayesian model to develop non-stationary transition probabilities describing uncertain climate states that reflect the new information that is available as the climate transitions.

Space is indeed a challenge. We have reframed the introduction substantially to focus in more detail on adaptive approaches in water resources planning, and also discuss implications for infrastructure investments for climate change adaptation broadly. We briefly mention applications to other fields on lines 63-66:

“Note that while this study focuses on water supply infrastructure, the challenge of characterizing learning about climate uncertainty to enable adaptive planning

has been highlighted in a range of other disciplines (see, for example Yousefpour 2012 in forest management).”

We have also changed the first sentence of the discussion section to more clearly emphasize this contribution:

“We develop a method that integrates iterative Bayesian learning about climate uncertainty into a multi-stage stochastic infrastructure planning model in order to address a critical limitation of adaptive infrastructure planning in both water supply and other domains: estimating upfront how much planners can expect to learn about climate change in the future and therefore whether adaptive approaches are likely to be reliable and cost effective.”

Reviewer #5 (Remarks to the Author):

- What are the major claims of the paper?

The paper is a case study on using flexible adaptation hydraulic structures for water resources projects. The study focuses on a region of Kenya where water availability is a challenge, and where a dam was proposed to be built. Based on a previous study indicating the costs of building a small or large dam, the authors propose to use a stochastic dynamic programming method to sequentially optimize and determine if the dams should be raised or not depending on climate evolution, remaining dam life and cost to upgrade. The authors state that this method could be used by practitioners to optimize dam costs and better manage climate change risk to water infrastructure.

Thank you for your review of our paper. We agree that flexible water infrastructure design is a focus of the paper. However, we see the key methodological contribution as: the novel method for quantifying opportunities to learn about climate change uncertainty in the future and using that to assess whether a flexible infrastructure approach is worthwhile. This is especially important for infrastructure development, where irreversible decisions have to be made up front and will have impacts for decades in the future.

We have substantially reframed the introduction, and as noted in the response to reviewer 4, we have also changed the first sentence of the discussion section to more clearly emphasize this contribution:

“We develop a method that integrates iterative Bayesian learning about climate uncertainty into a multi-stage stochastic infrastructure planning model in order to address a critical limitation of adaptive infrastructure planning in both water supply and other domains: estimating upfront how much planners can expect to

learn about climate change in the future and therefore whether adaptive approaches are likely to be reliable and cost effective.”

- Are the claims novel? If not, please identify the major papers that compromise novelty

Although the implementation using SDP for this purpose is novel to me, the idea of rethinking structural analysis and also operational adaptation (which is not dealt with in this paper) of dams is not new:

Ehsani, N., Vörösmarty, C.J., Fekete, B.M. and Stakhiv, E.Z., 2017. Reservoir operations under climate change: storage capacity options to mitigate risk. *Journal of Hydrology*, 555, pp.435-446.

Watts, R.J., Richter, B.D., Opperman, J.J. and Bowmer, K.H., 2011. Dam reoperation in an era of climate change. *Marine and Freshwater Research*, 62(3), pp.321-327.

Cherry, J.E., Knapp, C., Trainor, S., Ray, A.J., Tedesche, M. and Walker, S., 2017. Planning for climate change impacts on hydropower in the Far North. *Hydrology and Earth System Sciences*, 21(1), pp.133-151.

Cervigni, R., Liden, R., Neumann, J.E., Strzepek, K.M. (Eds.), 2015. Enhancing the Climate Resilience of Africa's Infrastructure: The Power and Water Sectors. The World Bank.

We agree that many previous studies have addressed adaptive management of dams as a way to mitigate the impact of climate change. Our contribution is differentiated from these previous studies in several ways.

First, our study focuses on managing the impacts of uncertainty in climate change adaptation, which makes it difficult to know if and when to adapt to climate change. Previous work has illustrated the difficulty in knowing whether recent trends in streamflow are a result of climate change or short-term variability and therefore whether they are predictive of future trends (Robinson and Herman 2018). How then, are planners to know when and whether to implement expensive changes? The studies above have a different focus. Ehsani et al. 2017, for example, provide a nice assessment of whether new dam capacity can mitigate the impacts of climate change, but does not address the question of how we will know these new dams are necessary and when to build them. Similarly, Watts et al. 2011 provide a helpful discussion of the options for adapting dams to climate change, but does not provide guidance on deciding if and when these changes will be necessary.

Second, we provide a method for estimating opportunities to learn about climate change uncertainty in the future. Adaptive management requires an ability to learn or have more information in the future that allows planners to trigger a change in course (Pahl-Wostl 2007). However, existing methods either assume perfect information about the future (as in Cervigni et al 2015) which can substantially overestimate the reliability of adaptive approaches. Other approaches rely on thresholds or tipping points which are either unspecified or set a priori without knowing whether they are the most appropriate (Haasnoot et al. 2013). Our Bayesian framework provides a formal quantification of

learning in the future. By applying the Bayesian uncertainty analysis to every possible observation of future climate, we characterize nonstationary stochastic transitions in the SDP that account for what will have been learned if a certain future climate comes to pass. Doing this allows us 1) to develop strategies that identify the optimal time to change strategy (if at all) based on what we will have learned and 2) decide upfront if investing in a flexible approach is worthwhile at all. It is the integration of the Bayesian analysis with the SDP that is the key contribution. Cherry et al. 2017 listed above in fact highlights the need to reduce uncertainty in the future as more observations are available and calls for new statistical approaches; here we develop a new statistical approach for reducing uncertainty and demonstrate its importance in infrastructure planning.

Third, we focus on new infrastructure development rather than changing operations of existing infrastructure. We have chosen this focus because assessing opportunities to learn in the future is most important for new infrastructure development; large scale capital investments require planners to decide upfront whether new infrastructure development should take a static or flexible approach. For example, a larger dam cannot be reduced in the future if it's not needed, and a smaller dam will never be able to take advantage of the full economies of scale of a larger dam later. The studies above do not focus on the development of new infrastructure, with the exception of Cervigni et al. 2015, which while it has a qualitative discussion of adaptive management, does not quantitatively assess flexible infrastructure design.

We have substantially reframed the introduction to make this focus more clear. We have also added more citations to existing work in flexible planning under climate change uncertainty to clarify our contribution. We have also made several edits to the methods and results section, listed above in the response to reviewer 4, to clarify the methodological contribution.

Pahl-Wostl, C. (2007). Transitions towards adaptive management of water facing climate and global change. *Water Resources Management*, 49–62.

Haasnoot, M., Kwakkel, J. H., Walker, W. E., & ter Maat, J. (2013). Dynamic adaptive policy pathways: A method for crafting robust decisions for a deeply uncertain world. *Global Environmental Change*, 23(2), 485–498.

Robinson, B., & Herman, J. D. (2018). A framework for testing dynamic classification of vulnerable scenarios in ensemble water supply projections.

- Will the paper be of interest to others in the field?

I think there will be interest from a small group of researchers and practitioners, but they are unlikely to try and find this information in a *Nature Communications Journal*. I

believe this work is best suited for a more specific, technical journal such as Journal of hydrologic engineering or Water Resources Management.

In addition to the points made above about the novelty and impact of the method, we also highlight that the method developed could be extended to the development of new infrastructure in many different domains (eg energy) that are impacted by uncertainty in climate change projections. UNEP estimates that the cost of climate change adaptation investments in the developing world may reach \$500 billion per year by 2050. Given the scarcity of capital for these investments, targeting investment resources effectively and efficiently is of utmost importance in order to protect a broader range of vulnerable communities. Our method provides a way to identify upfront places where more incremental, less expensive infrastructure projects are likely to be reliable.

We also note that one of the reasons for Nature Communications as a choice of journal rather than other Nature journals is its focus on publishing advances of importance to specialists within a disciplinary field.

- Will the paper influence thinking in the field?

I believe that it is a nice addition (in implementation and methodology) for specialists in this field, but is unlikely to influence the general mindset. There are too many hypotheses related to this case study to justify the generalization to other sites and uses (see comments below on the weather generator component). Furthermore, it is likely that adapting operation rules according to the changes in climate will help mitigate a certain amount of the risk.

Please see notes above making the contribution more clear, and response to the issue of generalizability below.

- Are the claims convincing? If not, what further evidence is needed?

The general premise is good and convincing, but some details make the results less conclusive to me, and especially not generalizable. For example, the methodology depends on a weather generator to simulate trajectories. The block bootstrapping method used here does not preserve autocorrelations on longer timescales (i.e. yearly) due to natural variability, thus averaging-out some of the variability that is expected to occur on longer timescales. Also, the monthly timestep is adequate for this volumetric study, but in no case could it be used for most systems which are managed on a daily or weekly time step. At these timescales, other interactions must be taken into account which this study fails to do. For example, extreme precipitation might drive the need for a larger reservoir, but if all we have is an increase of 10% in monthly precipitation, then this extreme event will not be taken into account. These are cases where management rules can be updated to adapt to climate change. I understand that the authors have a specific case in mind, and that the adaptations could still be used to further reduce dam size during initial sizing of the dams, but as it stands it is difficult for me to suggest that this paper is of broad enough interest for publication in a Nature journal.

We agree that using a monthly time-step and the choice of stochastic weather generator would not be appropriate for other applications, for example, in flood management where extreme precipitation events are important. We agree, as you note below, that the approach is sufficient for this application. Moreover, we again emphasize that the key contribution is the development and linkage of the Bayesian uncertainty analysis to the SDP. This part is indeed generalizable to many other studies. Other studies could then choose an appropriate water resource system model (including a stochastic weather generator), in order to translate long term climate states to infrastructure impacts in whatever way is most appropriate for the problem at hand. We therefore see the overall framework as generalizable and able to be tailored to a wide range of infrastructure planning problems.

We clarify this at the beginning of the methods section on lines 497-504:

“We note that the integration of the Bayesian statistical model with the SDP is the key methodological contribution and designed to be generalizable to many other domains. However, we demonstrate this on an example from water supply and therefore use a relatively simple water system model (comprised of the stochastic weather generator and infrastructure operations, described below) in this particular application. Future applications could tailor the water resource system model to the application at hand.”

• Are there other experiments that would strengthen the paper further? How much would they improve it, and how difficult are they likely to be?

Some experiments that could be done would be to use an SDP to re-optimize the management rules in time, and condition their work on that. Also, smaller reservoirs with more reactive times would be useful for the broader community. These additions would probably be difficult enough to put in place, so this is why I recommend the authors submit their work to a journal where their (very nice) incremental addition can be better positioned.

We acknowledge that the choice of fixed reservoir operations in time is a limitation and that including adaptive operating rules would likely further reduce the need for large additions of reservoir capacity. We are in fact currently planning a follow on study looking at the tradeoffs and interactions between adaptive reservoir operations and flexible infrastructure design. However, the main focus here is the methodological contribution around learning as noted above, and we think focusing the paper on that rather than adding more nuance to the application is more likely to clearly communicate the contribution.

We have added a sentence on lines 720-723 of the method section acknowledging this limitation:

“We acknowledge that assuming reservoir operations that are fixed in time is a limitation given that adaptive reservoir operations would likely reduce the need

for additional capacity; future work could optimize the reservoir operations to each climate state.”

- Are the claims appropriately discussed in the context of previous literature?

Yes, the authors did a good job in their literature review and placed their work in context.

- If the manuscript is unacceptable in its present form, does the study seem sufficiently promising that the authors should be encouraged to consider a resubmission in the future?

As per my points above, I think the subject is good, the authors did a good job for their case study, but that it is simply too narrow of a research field for this journal.

Code and Data:

Available at: https://github.com/sfletcher23/Fletcher_2019_Learning_Climate

Reviewers' comments:

Reviewer #1 (Remarks to the Author):

The authors have satisfactorily addressed my comments.

Reviewer #4 (Remarks to the Author):

The authors improved the explanation of their methodology. However, the authors did not understand my main objection to their methodology correctly, and I think it remains valid.

I did not suggest that the authors use the median of the distribution at time $t=1$ as observation at time $t=2$ etc.

I try to explain again in other words:

In simple terms, the fundamental flaw in the procedure is that essentially the same information is used in several Bayesian update steps (Equation 1 of the Methods section). That is about the worst you can do in Bayesian statistics. The information that you use in sequential Bayesian update steps needs to be new information. But in the proposed procedure, the information that is used to update the distribution at time $t=1$ is (essentially) the same information that is used to update the distribution at time $t=2$ and $t=3$ and time $t=4$ etc.

To make the point clear, let us focus on Figure 1, the panel on the bottom left (the climate model projections). The different climate models are all based on a set of historical observations, called ObsHist, and some (varying) knowledge in numerical modelling and physics, called ExpertKnowledge. These observations and this knowledge (information) is available at time $t=0$.

In the proposed uncertainty quantification, the different Bayesian updates, at different times, are all based on the climate model projections (performed at $t=0$) and the original historical data (available at time 2000). But the information which is contained in the climate model projections (and, possibly, the weighting of the projections using historical observations), is basically ObsHist plus ExpertKnowledge. This information is essentially the same at all times, even though you consider climate projections at different times. The time here is a pure label, it does not mean that including more and more points in time really adds any new information. The considered climate model projections do not include more real information than ObsHist plus ExpertKnowledge (let us neglect natural variability, that is not the point here). This can also be seen from the fact that the transition probability distributions will depend on how many times you perform the Bayesian updating process, but you do not really use any more information, regardless whether you update 5 times, 10 times, or hundred times.

The climate projections, at time $t=1$, are basically just a reformulated version of the data ObsHist plus ExpertKnowledge. That is fine, but then the climate model data at $t=2$ is essentially the same, just a reformulated version of ObsHist plus ExpertKnowledge. It is not really new information! Think of the climate models as linear regression models (which they could be). It cannot be new information! Obviously, new information on the future state of the climate at time $t=2$ will only become available at time $t=2$.

So what you do in your procedure is some sort of magic: you claim to have new data (new information) available at time $t=2010$, $t=2020$, $t=2030$, etc., always using essentially the same information, namely the one we have in the year $t=2000$ (I appreciate that you do not just use one single climate projection, or a discrete set of paths/projections to generate the probability distributions shown in Figure 3, but still, there is no real new information contained in the climate projections used at different future times). Only because the climate model projection for the year

2030 bears the label "2030" does not mean that this is new information which becomes available in the year 2030. As I said in my previous comments, it is okay to use the climate model projections in the way that you construct a posterior distribution at ONE future time (say 2070), using historical observations (data and climate models from the year 2000). But you cannot use the same data several times, in sequential Bayesian updating steps, pretending that the climate model data at different future times is actually new information that will be observed in the future. In the future, new data will become available, but we don't know what it is. Only once the new information will become available can you do another, additional Bayesian update step, using this newly available information.

Two related, more specific point:

I think I understand now how you use what you, in the revised manuscript, call "virtual observations", but the explanation is not very clear. Please, in the Methods section, lines 550-552, denote the virtual observations by a letter, and include them in Equation 1 in an appropriate way, so that it can be understood where they (literally) enter the equation. Also, include both the virtual observations as well as the climate model data in the schematic, Figure 1, and make clear that the "virtual observations" are different from the climate model data. These virtual observations and their ranges seem completely arbitrary, another problematic aspect of the methodology.

Please write a complete, correct formula for the transition probabilities $p(s(t+1)|s(t),a(t))$, clearly differentiating between $s(t)$, the climate model projection data of the different models at different times, and the "virtual observations". The transformation of relative to absolute values is not so important here.

I still think, for the above reasons, that the sequential, multiple update procedure in the present article is problematic, and I have serious concerns regarding the methodology.

Reviewer #5 (Remarks to the Author):

I have reviewed the revised version of the manuscript, and I think the authors did a good job in (1) addressing my main comments about the scope and methodology and (2) respond to my inquiries about the choice of journal. After consideration, I think the paper should be published. I have not noted any methodological flaw (except for the hypothesis about fixed optimization rules, but the authors responded adequately), the results are convincing and I think that the paper should move forward in the publication process.

Reviewer #4 (Remarks to the Author):

The authors improved the explanation of their methodology. However, the authors did not understand my main objection to their methodology correctly, and I think it remains valid.

I did not suggest that the authors use the median of the distribution at time $t=1$ as observation at time $t=2$ etc.

I try to explain again in other words:

In simple terms, the fundamental flaw in the procedure is that essentially the same information is used in several Bayesian update steps (Equation 1 of the Methods section). That is about the worst you can do in Bayesian statistics. The information that you use in sequential Bayesian update steps needs to be new information. But in the proposed procedure, the information that is used to update the distribution at time $t=1$ is (essentially) the same information that is used to update the distribution at time $t=2$ and $t=3$ and time $t=4$ etc.

To make the point clear, let us focus on Figure 1, the panel on the bottom left (the climate model projections). The different climate models are all based on a set of historical observations, called ObsHist, and some (varying) knowledge in numerical modelling and physics, called ExpertKnowledge. These observations and this knowledge (information) is available at time $t=0$.

In the proposed uncertainty quantification, the different Bayesian updates, at different times, are all based on the climate model projections (performed at $t=0$) and the original historical data (available at time 2000). But the information which is contained in the climate model projections (and, possibly, the weighting of the projections using historical observations), is basically ObsHist plus ExpertKnowledge. This information is essentially the same at all times, even though you consider climate projections at different times. The time here is a pure label, it does not mean that including more and more points in time really adds any new information. The considered climate model projections do not include more real information than ObsHist plus ExpertKnowledge (let us neglect natural variability, that is not the point here). This can also be seen from the fact that the transition probability distributions will depend on how many times you perform the Bayesian updating process, but you do not really use any more information, regardless whether you update 5 times, 10 times, or hundred times.

The climate projections, at time $t=1$, are basically just a reformulated version of the data ObsHist plus ExpertKnowledge. That is fine, but then the climate model data at $t=2$ is essentially the same, just a reformulated version of ObsHist plus ExpertKnowledge. It is not really new information! Think of the climate models as linear regression models (which they could be). It cannot be new information! Obviously, new information on the future state of the climate at time $t=2$ will only become available at time $t=2$.

So what you do in your procedure is some sort of magic: you claim to have new data (new information) available at time $t=2010$, $t=2020$, $t=2030$, etc., always using essentially the same information, namely the one we have in the year $t=2000$ (I appreciate that you do not just use one single climate projection, or a discrete set of paths/projections to generate the probability distributions shown in Figure 3, but still, there is no real new information contained in the climate projections used at different future times). Only because the climate model projection for the year 2030 bears the label "2030" does not mean that this is new information which becomes available in the year 2030. As I said in my previous comments, it is okay to use the climate model projections in the way that you construct a posterior distribution at ONE future time (say 2070), using historical observations (data and climate models from the year 2000). But you cannot use the same data several times, in sequential Bayesian updating steps, pretending that the climate model data at different future times is actually new information that will be observed in the future. In the future, new data will become available, but we don't know what it is. Only once the new information will become available can you do another, additional Bayesian update step, using this newly available information.

Response to Reviewer 4 general comments:

Thank you for clarifying your comments about the methodology. We misinterpreted your previous comments and believe we now understand your concern. We still see it as a misunderstanding due to a lack of clarity in our discussion of the virtual observations and their connection to the SDP state space, rather than an error in the approach.

We absolutely agree that it would be invalid to sequentially update using only a reformulation of the climate model projections without adding any additional information (i.e. model projections at 2050 instead of 2030 are still developed from the same model dynamics and historical data and therefore do not contain new information). This is not our approach. We are in fact using different information each time we estimate the Bayesian model: a "virtual" climate observation that does not come from the climate model projections, and rather comes from the SDP system state, following common practice in stochastic control. The SDP state space represents all the future climates we may reach, and the SDP policy describes what we should do if we reach any specific state. By developing virtual observations that correspond to the change in SDP climate state, and using them as observations in the Bayesian model, we can develop the necessary SDP policies that account for what we will have learned if we reach a particular state in the future. This is in contrast to common practice in our field which inherently assumes that our uncertainty in future climate change projections will remain as uncertain in 50 years as it is today.

Let us explain in detail:

You are right indeed that we won't have more information about the state of the climate at $t=2$ until $t=2$. But, as you say, once we get to $t=2$, we can then use the new observed temperature data to estimate a new distribution. Let's say, just hypothetically, that the temperature change value we observe at $t=2$ is 1 degree C. We know that 1 degree C of change is a possible value that we might observe in $t=2$. And we have all the necessary statistical mechanics to estimate a new uncertainty distribution for $t=3$ if we were to observe 1 degree C of change in $t=2$. So we can estimate the new uncertainty distribution for $t=3$ conditioned on observing 1 degree C of change in $t=2$.

Of course we may not observe 1 degree C of change in $t=2$! We might observe 0.4 degrees C of change. We might also observe 1.65 degrees C of change. So we repeat the procedure for those values as well. We define a comprehensive set of possible values we might observe – the set of virtual future observations, based on the climate system states in the SDP – and we repeat the procedure for each of them. In this way, we're saying: here's how the distribution forecasting for $t=3$ will change if X amount of change is observed in $t=2$, for all possible values of X . This is used to characterize the SDP transition probabilities, which require a probability of reaching state a in $t=3$ given that we reached state b in $t=2$.

That is the key element: the change in the SDP temperature state **IS** the virtual temperature observation. We haven't observed a temperature in $t=2$ yet, but we have an SDP that lays out all the possible values we might observe through its system state space. And repeating the Bayesian analysis for each unique state allows us to say what we will have learned if we reach (and therefore observe) that state in the SDP. This provides unique transition probabilities for each state, that take into account the additional information we will have if that state is reached. The result of the SDP is a policy, or optimal action as a function of the system state, that accounts for predicted transitions from that state. This is a form of stochastic closed-loop control, where new information about the system feeds back into updated estimates for system state transitions over time.

There are many examples in the literature that use an SDP formulation in which the transition probabilities are unique for each state and reflect the new information that will be known when that state is reached. McDonald-Madden et al. (2012) assess optimal timing of species relocation under climate change uncertainty using SDP. One of their SDP state variables is the degree of belief in climate impact, and the transition probabilities for that state are estimated using Bayes' with the current state treated as an observation (see equation 4 in the SI). Fackler et al. (2013) present a wildlife management example using SDP in which unique transition probabilities are estimated by incorporating the current state variable as a parameter in a Poisson distribution (see equation 11). Neither of these studies uses actually observed data to develop or update probability distributions describing the state transitions; they utilize the system state as an observation. Our approach of using virtual future observations based on the current SDP system state to develop unique updated transition probabilities with a Bayesian model is analogous to these.

McDonald-Madden, E., Runge, M. C., Possingham, H. P., & Martin, T. G. (2011). Optimal timing for managed relocation of species faced with climate change. *Nature Climate Change*, 1(8), 261–265. <https://doi.org/10.1038/nclimate1170>

Fackler, P. L., Marescot, L., Chapron, G., Chad, I., Duchamp, C., Marboutin, E., & Gimenez, O. (2013). Complex decisions made simple : a primer on stochastic dynamic programming, 872–884. <https://doi.org/10.1111/2041-210X.12082>

We realize now, thanks to your comments, that this was not made clear mathematically. We have made the follow updates to the paper to clarify:

1) We make a number of changes to the equations in the methods section including:

- We have updated equation 1 and split it into two versions: one (now equation 1) in which historical observed data is used to condition which we use in $t=1$, and a second (equation 2) in which virtual future observations are used to condition instead which we use in $t > 1$. This demonstrates more clearly that each estimate of change in the next period is based on a unique virtual observation. This appears in lines 611-635.
- We have added a new subsection in the methods that derives the transition probabilities from the estimated distributions and virtual observations in the Bayesian model. This subsection appears in lines 649-666.
- A number of notation changes in the equations have been made to make the SDP notation consistent with the Bayesian model.

2) In the paragraph in the introduction where we first introduce the virtual climate observations, we make their connection to the SDP state change explicit in lines 80-83: “These updated uncertainty estimates characterize the transition probabilities in a non-stationary stochastic dynamic program (SDP); each possible in SDP climate state is equivalent to a virtual climate observation.”

3) We add a simple analogy in the introduction on lines 90-108 for readers to get a better intuitive sense for the virtual observations and their connection to SDP and flexible planning:

”While we do not know today what observations we will see in the future, we can develop policies today for what we will do if a certain observation comes to pass in the future. As an everyday analogy, say we are planning to host a party next week. Our friends are slow to respond to our invitation, and we do not yet know how many people will attend. Therefore, we do not know if our current supply of drinks is sufficient. If we make a final decision today about whether to buy more drinks, we risk unhappy guests if our supply is insufficient or overspending if we buy too much. We can, however, calculate the maximum possible number of guests and assess whether our current supply of drinks is sufficient. If it is

sufficient in the maximum case, we can go about our week reassured. If it is not, we can make a plan to reevaluate the responses the day before the party and save time in our day to go to the store for more drinks. We will do this if the expected demand for drinks in light of our updated information exceeds our supply, and in fact we can decide today what number of day-ahead guest responses would prompt us to buy more drinks. In this way, we are developing policies for future actions (going to the store; adding water supply capacity) based on the information from virtual future observations (day-ahead guest responses; temperature and precipitation change) in order to determine whether we should build flexibility into our plan today (saving time for a future errand; choosing a flexible dam design).”

4) We add the above references to the description of SDP in the methods section in lines 674-677:

“This is analogous to existing approaches in ecology, which have defined SDP transition probabilities with probability density functions that include the current system state as an input (McDonald-Madden2011, Fackler2013).”

Two related, more specific point:

I think I understand now how you use what you, in the revised manuscript, call "virtual observations", but the explanation is not very clear. Please, in the Methods section, lines 550-552, denote the virtual observations by a letter, and include them in Equation 1 in an appropriate way, so that it can be understood where they (literally) enter the equation.

Done – see note above. Virtual observations are now referred to as $\Delta V_{t,i}$, where t denotes the time period and i is an index from 1 to N (total number of virtual observations). This is defined on lines 576-579.

Also, include both the virtual observations as well as the climate model data in the schematic, Figure 1, and make clear that the "virtual observations" are different from the climate model data.

Done. We also connected this to the new mathematical notation in the methods section.

These virtual observations and their ranges seem completely arbitrary, another problematic aspect of the methodology.

The range and discretization of the virtual observations are based on the range and discretization of the SDP climate state space. As is best practice for defining an SDP state space, we experimented with the discretization until we had one granular enough to reasonably approximate a continuous state space. We also confirmed that it was granular enough that adjacent virtual observations lead to similar updated transition probabilities. For example, observing 1.00 degrees of change leads to a probability distribution that is similar to that based on observing 1.05 degrees of change; therefore,

it is not necessary to include 1.01 as well. We also made sure the range was wide enough to exceed the range predicted by the ensemble of CMIP5 models. This was clarified in the text by adding on lines 582-590 of the methods section:

“These were chosen in order to be comprehensive of all potential future climate states. Therefore, they must 1) be granular enough that adjacent observations result in similar distributions and therefore approximate a continuous set of observations and 2) span a range that exceeds the full range of change predicted by models (i.e. a range of 0 to 1.5 °C per 20-years is equivalent to 0 to 7.5 °C of change after 100 years; the CMIP5 ensemble projections a temperature change in the range of 2 to 4°C by 2100, fitting well within the range resulting from the virtual observations).”

Please write a complete, correct formula for the transition probabilities $p(s(t+1)|s(t),a(t))$, clearly differentiating between $s(t)$, the climate model projection data of the different models at different times, and the "virtual observations". The transformation of relative to absolute values is not so important here.

We have added a subsection in the methods (lines XX) that derives the transition probabilities from the Bayesian model, including the virtual observations. We provide a closed-form equation for the joint distribution of the SDP states based on the virtual observations in equation 8. We apply Monte Carlo simulation to get the conditional (transition) probabilities from the joint (we found this simpler than integrating the joint); we provide our algorithm for this approach.

We took out the paragraph that had previously described the transformation of relative to absolute (this is now contained in the quantitative derivation).

I still think, for the above reasons, that the sequential, multiple update procedure in the present article is problematic, and I have serious concerns regarding the methodology. We hope our response has grounded our approach in existing SDP literature and clarified that we are not sequentially updating using the same information. Rather, we estimate the Bayesian model many times for unique virtual climate observations, each of which correspond to a change in the SDP climate state. While we haven't observed these observations today, if we get to a state X in the SDP at time t, then X will be an observation. So we can use this to develop SDP policies for what we should do if we reach state X in the future that take into account what we will have learned if we reach state X.

REVIEWERS' COMMENTS:

Reviewer #4 (Remarks to the Author):

The authors have improved the explanation and discussion of their methodology, and the manuscript can now be accepted for publication.